# TestGenEval: A Real World Unit Test Generation and Test Completion Benchmark

**Kush Jain**[1,2]**, Gabriel Synnaeve** [2]**, Baptiste Rozière**[2]
[1] Carnegie Mellon University, [2] FAIR, Meta AI
{kdjain}@andrew.cmu.edu

## Abstract

Code generation models can help improve many common software tasks ranging from code completion to defect prediction. Most of the existing benchmarks for code generation LLMs focus on code authoring or code completion. Surprisingly, there has been far less effort dedicated to benchmarking software testing, despite the strong correlation between well-tested software and effective bug detection. To address this gap, we create and release TestGenEval, a large-scale benchmark to measure test generation performance. Based on SWEBench, TestGenEval comprises 68,647 tests from 1,210 code and test file pairs across 11 well-maintained Python repositories. It covers initial tests authoring, test suite completion, and code coverage improvements. Test authoring simulates the process of a developer writing a test suite from scratch, while test completion mimics the scenario where a developer aims to improve the coverage of an existing test suite. We evaluate several popular models, with sizes ranging from 7B to 405B parameters. Our detailed analysis highlights TestGenEval's contribution to a comprehensive evaluation of test generation performance. In particular, models struggle to generate high-coverage test suites, with the best model, GPT-4o, achieving an average coverage of only 35.2%. This is primarily due to models struggling to reason about execution, and their frequent assertion errors when addressing complex code paths.

## 1 Introduction

Software testing is a critical component of the software development process. A high-quality test suite can be instrumental in finding inconsistencies between a system's specifications and its implementation. Test suites ideally execute all code paths (high code coverage), and catch regressions in the code under test that a developer might introduce (high mutation score) (Fraser & Arcuri, 2013; Veloso & Hora, 2022). However, writing high quality tests can be time-consuming (Beller et al., 2015a;b) and is often either partially or entirely neglected.

As a result there has been extensive research on automated test generation, including both classical (Fraser & Arcuri, 2011; Brandt & Zaidman, 2022; Baldoni et al., 2018) and neural-based methods (Dinella et al., 2022; Watson et al., 2020; Touvron et al., 2023b; OpenAI, 2023). However, despite this growing area of research, unit test generation benchmarks remain limited in size and scope (Chen et al., 2021; Bhatia et al., 2024). While existing benchmarks capture test generation abilities on simple, typically self-contained programs, there is an absence of large scale test generation benchmarks corresponding to real world use cases. Other related benchmarks (Jain et al., 2024) report performance on adjacent tasks such as generating equivalence tests rather than standard unit tests, which also differs from the real world use case. Current benchmarks report pass@k, with few reporting code coverage and none reporting mutation score, despite mutation score being most correlated with real fault detection (Just et al., 2014; Papadakis et al., 2018).

Existing benchmarks also do not measure test completion capabilities, despite many code completion benchmarks existing (Liu et al., 2023; Zhuo et al., 2024). Test completion can be used to add tests to an already existing unit test file and improve overall coverage. This is important for IDE auto-completion features, where given a part of a test file an the code under test, the goal is to add more tests. Test completion is also measured by many state of the art software testing models (Rao

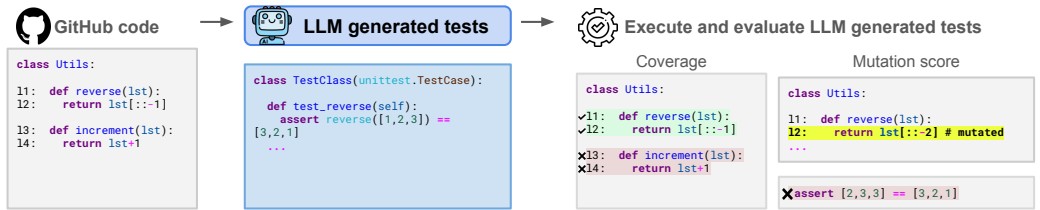

Figure 1: An overview of TESTGENEVAL. We start with a GitHub code file and generate a test suite with an LLM. Then we execute the generated test suite and measure the proportion of lines in the code file that are executed (code coverage). We also inject synthetic bugs into the code and measure the proportion of synthetic bugs detected by the generated test suite (mutation score).

et al., 2023; Nie et al., 2023; Dinella et al., 2022; Tufano et al., 2020), yet a benchmark that measures test completion doesn't exist.

Motivated by this, we introduce TESTGENEVAL with two tasks 1) full file unit test generation and 2) test completion (see Section 2.2 for more details). Our benchmark consists of real world projects, with each source file containing an average 1,157 lines of code (LOC) and each test file containing an average of 943 LOC. TESTGENEVAL consists of 68,647 tests from 1,210 unique code-tests file pairs. For fast iteration in low-compute settings, we also provide a smaller version of the benchmark TEST-GENEVALLITE, which approximates all the metrics computed in TESTGENEVAL (see Appendix E for more details). TESTGENEVALLITE includes 160 code-tests file pairs, file unit test generation, and test completion tasks. It was sampled to be representative of the full TESTGENEVAL: the repositories and other statistics are similar in TESTGENEVALLITE and TESTGENEVAL (see Appendix A for more details and Appendix G for statistical significance tests).

We find that models struggle to generate high quality test suites (Section 3.1). The best performing model—GPT-4o—has an average coverage of 35.2% and a mutation score of 18.8%. Generating tests for large scale projects is significantly harder than generating tests for self-contained problems; this is reflected by significantly lower scores compared to existing benchmarks such as TestEval (Wang et al., 2024), where top models achieve nearly 100% line coverage. Test completion (Section 3.2) is significantly easier than test generation, reflected by the high pass@5 rates of the best performing models. However, models struggle to add coverage to a complete test suite, with top models adding less than 1% coverage when generating the last test for an existing file.

We perform extensive quantitative and qualitative analysis of all results (Section 4). We measure correlation between TESTGENEVAL and other popular benchmarks, the correlation between the test generation and test completion tasks, and the correlation between models for each task. We also perform an analysis of errors, and measure the effects of sampling more and context size on TEST-GENEVAL performance (Section 4.1). We also examine cases where TESTGENEVAL can discriminate between highly performing models (Section 4.2). More analysis can be found in Appendix F.

We provide all the code for our benchmark at `https://figshare.com/s/51171ae97cd21d233d4f`, including detailed instructions on how to run our benchmark, and even extend it. We also provide a website with all model generations for TESTGENEVAL. We hope that this will enable the community to use TESTGENEVAL and further build upon our work.

Our contribution are as follows:

- We release a benchmark for partial and full test suite generation on a realistic set of 1,210 snippets in 11 repositories. We use coverage and mutation score metrics to evaluate the value of the generated test suites

- We evaluate various prominent open and closed-source code generation models on our benchmark, and show that, for large scale repositories, models struggle to generate high coverage test suites

- We release docker images allowing to easily run code from these 11 repositories and evaluate scores on our benchmark

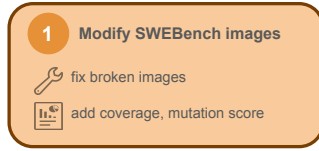 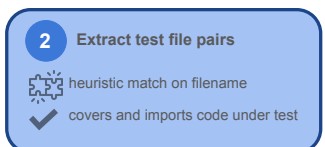 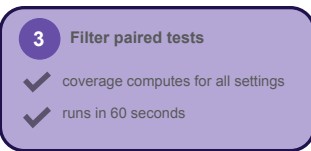

Figure 2: A pipeline describing the creation of TESTGENEVAL. We start with docker images of the SWEBench dataset and instrument them with coverage and mutation score dependencies. Then we extract code test file pairs by performing a heuristic match on filename. Finally, we filter out tests that don't have coverage on the code under test and run in 60 seconds.

## 2 TESTGENEVAL

TESTGENEVAL consists of 1,210 code test file pairs from 11 large, well-maintained repositories (3,523-78,287 stars). We use these file pairs to construct two testing tasks: 1) unit test completion for the first, last and additional tests and 2) full file unit test generation. Our benchmark is easy to run and extend, as we have docker containers for each version of each repository with coverage and mutation testing dependencies installed. For both tasks we use execution based metrics, including pass@1, pass@5 along with code coverage improvement, and mutation score improvement compared to the gold (human written) tests. Code and test files in TESTGENEVAL are long in length (on average 782 LOC per code file and 677 LOC per test file) and high coverage (median coverage of 60.4%). Detailed statistics of TESTGENEVAL can be found in Appendix A.

### 2.1 BENCHMARK CONSTRUCTION

We construct TESTGENEVAL by adapting the SWEBench dataset to software testing tasks. Figure 2 shows the full set of steps to construct TESTGENEVAL. We apply these exact steps to SWEBench-Lite to construct TESTGENEVALLITE.

**Modify SWEBench images:** We start with the docker images provided by SWEBench [1]. We first modify the images that did not build manually, by installing appropriate dependencies and modifying requirements files so every test suite executes. Next, we install both coverage and mutation testing dependencies and modify the test commands to run with coverage instrumentation.

**Extract test file pairs:** After we have the execution environment for each pull request version built, we next extract code test file pairs from the code and test files run in the PR. Specifically, we extract code test file pairs by performing a heuristic match on filenames, and filtering out pull requests that do not meet our heuristic. With each file pair we perform program analysis on the test file to extract the first test, last test as context for our test completion settings. We later validate for each file pair that the gold tests cover some part of the code under test.

**Filter paired tests:** Next, we run code coverage for all the gold test settings (first, last, and extra) to filter out repositories where contexts were extracted incorrectly or partial test files do not run. We also filter out tests that take longer than 60 seconds to run to ensure TESTGENEVAL runs efficiently.

### 2.2 TASKS

Figure 5 shows an example of both testing tasks. TESTGENEVAL consists of 2 separate tasks: test generation and test completion.

**Test generation:** The goal of test generation is to generate an entire test suite given a file under test. We provide the necessary inputs to the model as part of the prompt (see Appendix C for full details). Our test generation task aligns with real world unit testing; unit testing in practice involves writing tests for large code files in complex projects.

**Test completion:** The goal of test completion is to generate the next test in an existing test suite given an existing test suite and the file under test. Test completion is measured by many software

---

[1]https://github.com/aorwall/SWE-bench-docker

```
class Character:
  ...
  def level_up(self):
    self.level += 1

  def damage(self, health):
    self.health -= heath

# TestGenEval-Full: full test suite
    generation

def test_character_setup():
  ...

def test_character_levels_up():
  ...

def test_character_damage():
  ...
```

Figure 3: Full test suite generation

```
class Character:
  ...
  def level_up(self):
    self.level += 1

  def damage(self, health):
    self.health -= heath

# TestGenEval test prefix

def test_character_setup():
  ...

def test_character_levels_up():
  ...

# TestGenEval: test completion

def test_character_damage(self):
  ...
```

Figure 4: Test completion (last)

Figure 5: Two software testing tasks (and highlighted model generations). Full test suite generation requires knowledge of code under test setup, along with meaningful assert statements. Test completion requires understanding the code under test and current test to generate an additional test method (first, last, and extra test).

testing models, and can be applied to in IDE tools, however there is no benchmark for this task. The test completion task aligns with different stages in the development life cycle; first test completion mirrors a developer starting their test suite, last test completion mirrors the finishing of their test suite and extra test completion measures whether language models can add an additional test to a test suite a developer thinks is complete. This setup is in line with Rao et al. (2023), which models test generation at the method level.

## 2.3 PROPERTIES OF TESTGENEVAL:

Table 1: Comparison of TESTGENEVAL against existing test generation benchmarks. TEST-GENEVAL mirrors the real world task of unit test generation, including file level, human written tests. TESTGENEVAL includes both individual test generation and full test suite generation, and is the only benchmark to measure mutation score.

| Dataset | Size | File level | Human tests | Test suite | Mutation score |
|---------|------|-----------|-------------|------------|----------------|
| TESTEVAL | 210 | ✗ | ✓ | ✗ | ✗ |
| SWT-BENCH | 1762 | ✓ | ✓ | ✗ | ✗ |
| R2E | 246 | ✓ | ✗ | ✓ | ✗ |
| TESTGENEVAL | 1210 | ✓ | ✓ | ✓ | ✓ |

Table 1 outlines some of the properties that differentiate TESTGENEVAL from existing test generation benchmarks. We explain each property in depth.

**File level test generation:** Real world unit testing involves reasoning over complex files, generating tests for a given file under test. Unlike existing benchmarks that deal with a small, self contained programs, TESTGENEVAL includes code and tests from large scale, highly starred projects.

**Human-written tests:** Existing benchmarks such as R2E (Jain et al., 2024) measure the ability of an LLM to generate equivalence tests. While models that generate equivalence tests have the advantage of high coverage and ease of scaling to large amounts of data, they often look different from developer-written tests (Kambhamettu et al., 2022). Measuring the ability of LLMs to generate equivalence tests is an adjacent, but different task from measuring unit test generation capabilities. TESTGENEVAL is the first large-scale test-generation benchmark using human-written tests.

**Test suite generation:** The goal of unit test generation is to generate high-quality test suites, which is correlated to high coverage and mutation score for the code under test. However, existing methods such as TestEval (Wang et al., 2024) and SWT-Bench (Mündler et al., 2024) only measure the ability of an LLM to generate an individual test method rather than an entire test suite. We propose complementing individual test completion tasks with the broader test suite generation task.

**Mutation score:** Unlike existing benchmarks, we report mutation score in conjunction with coverage. The mutation score is empirically far more correlated with bug detection capabilities (Just et al., 2014; Papadakis et al., 2018) and much harder to hack; in order to achieve high mutation score tests must be able to discriminate non-buggy code from code with synthetic bugs introduced.

## 3    EVALUATION

We evaluate a selection of models on TESTGENEVAL to better understand how models of different sizes and families (see Table 2 for list, and Appendix D.1 for more details about model choices) perform at test suite generation and test completion. The HuggingFace checkpoints of all open source models used are also available at Appendix B. We prompt each model with the maximum context window size possible, otherwise truncate the starting tokens to fit the prompt in the context window.

We report results for all models in both the full test generation (Section 3.1) and test completion tasks (Section 3.2) on TESTGENEVAL. For test suite generation we report any pass@1 (if any of the tests in the generated test suite pass), all pass@1 (if the generated test suite passes), coverage (coverage of passing tests), and mutation score (proportion of synthetic bugs introduced to code caught by test suite). For test completion we report pass@1 and pass@5 (whether generated test passes), along with coverage improvement from adding the generated test. More detailed descriptions and our full set of metrics can be found at Appendix D.2. Our full set of results for TESTGENEVAL and results for TESTGENEVALLITE can be found in Appendix E. We also perform statistical significance tests and report 95% confidence intervals in Appendix G.

### 3.1    TEST GENERATION PERFORMANCE OF VARIOUS MODELS

Table 2: Full test suite generation. All results shown for temperature=0.2. Larger models generally perform better at test suite generation, however all models struggle to achieve high coverage and mutation score.

| Model | All Pass@1 | Any Pass@1 | Coverage | Mutation |
|---|---|---|---|---|
| **Small Models** | | | | |
| CodeLlama 7B | 3.2% | 4.1% | 1.2% | 0.5% |
| Gemma 9B | 3.5% | 42.1% | 20.2% | 9.0% |
| Llama 3.1 8B | 3.1% | 30.5% | 14.1% | 6.8% |
| **Medium Models** | | | | |
| DeepSeekCoder 16B | 23.0% | 63.7% | 28.2% | 12.1% |
| Gemma 27B | 7.4% | 57.7% | 30.1% | 14.6% |
| Codestral 22B | **26.8%** | 72.7% | 33.0% | 14.2% |
| **Large Models** | | | | |
| CodeLlama 70B | 14.0% | 22.7% | 7.0% | 2.5% |
| Llama 3.1 70B | 7.8% | 60.7% | 30.6% | 15.4% |
| **Flagship Models** | | | | |
| GPT-4o | 7.5% | 64.0% | **35.2%** | **18.8%** |
| Llama 3.1 405B | 17.7% | **73.1%** | 35.0% | 16.4% |

Table 2 shows any pass@1, all pass@1 coverage and mutation score for small, medium and large models. All three smaller models perform significantly worse than their larger counterparts, with

a large difference (42.6% for Llama 3.1 models and 15.6% for Gemma models in any pass@1 performance). All pass@1 penalizes how verbose a model is: DeepSeekCoder 16B on average generates a test suite with 106 lines of code and 16 methods, while Llama 3.1 70B generates an average of 324 lines of code and 36 methods. This intuitively makes sense, as the more verbose a model is, the more likely it is to generate a test with errors.

Coverage and mutation score remain low across all models. For coverage, GPT-4o performs the best, but still covers only 35.2% of the lines of code tested in TESTGENEVAL. The mutation score is even lower than the coverage, which implies that tests generated by these models cannot catch all induced bugs. Despite generating passing tests for a smaller proportion of programs, GPT-4o achieves higher coverage than Llama 3.1 405B, meaning that GPT-4o generates higher quality tests on average (also reflected by the higher mutation score of GPT-4o compared to Llama 3.1 405B).

## 3.2 TEST COMPLETION PERFORMANCE OF VARIOUS MODELS

Table 3: Pass rates for different models in first, and last settings. Pass@1 is for temperature=0.2, Pass@5 is for temperature=0.8. Codestral 22B outperforms all models across pass@1 and pass@5 (other than first test completion pass@5), solving between 60-70% of all tasks.

| Model | First | | | Last | | |
|---|---|---|---|---|---|---|
| | Pass@1 | Pass@5 | +Cov | Pass@1 | Pass@5 | +Cov |
| **Small Models** | | | | | | |
| CodeLlama 7B | 4.2% | 19.8% | 6.7% | 6.9% | 24.0% | 0.0% |
| Gemma 9B | 8.4% | 18.0% | 6.7% | 21.4% | 46.4% | 0.1% |
| Llama 3.1 8B | 14.4% | 33.3% | 12.8% | 32.0% | 54.3% | 0.1% |
| **Medium Models** | | | | | | |
| DeepSeekCoder 16B | 18.6% | 47.0% | 19.0% | 17.0% | 62.8% | 0.2% |
| Gemma 27B | 12.7% | 33.7% | 13.6% | 32.2% | 62.2% | 0.1% |
| Codestral 22B | **38.3%** | 61.7% | 24.0% | **50.4%** | **74.3%** | 0.4% |
| **Large Models** | | | | | | |
| CodeLlama 70B | 0.5% | 30.2% | 11.2% | 0.9% | 50.7% | 0.0% |
| Llama 3.1 70B | 19.3% | 46.4% | 18.7% | 35.0% | 61.9% | **0.5%** |
| **Flagship Models** | | | | | | |
| GPT-4o | 31.9% | **63.5%** | **26.9%** | 32.6% | 66.6% | **0.5%** |
| Llama 3.1 405B | 32.1% | 57.7% | 21.6% | 42.6% | 72.3% | 0.3% |

Table 3 shows pass@1, pass@5 and coverage improvement for first and last settings. Information on extra test completion can be found in Appendix E). Model performance generally increases as more of the test file is provided as context (extra pass@5 is higher than both last pass@5 and first pass@5). Similar to the full test setting, larger models tend to outperform smaller ones. An outlier is Llama 3.1 8B, with a significantly higher pass@5 in all settings than its smaller model counterparts. Codestral 22B performs the best at generating passing tests, with a pass@5 of 74.3% on the last test completion setting. Models face challenges in augmenting coverage for existing human-written test-suites, whereas they can more readily add coverage when no tests are initially present. In the final test completion setting, all models generate virtually no new coverage, primarily testing computation paths that have already been covered.

## 4 ANALYSIS

We perform a quantitative and qualitative analysis of all results. This includes correlation with other benchmarks, effects of samples on pass@k, effects of context window size along with a qualitative analysis of differentiating problems between Codestral, GPT-4o and Llama 405B. Our complete analysis can be found in Appendix F.

## 4.1 QUANTITATIVE ANALYSIS

We perform an analysis of model correlation with other benchmarks (Section 4.1.1) along with the effects of sampling more (Section 4.1.2) and increasing context window (Section 4.1.3) on model performance. Details on correlation between models and settings and common model errors can be found in Appendix F.2 and Appendix F.5 respectively.

### 4.1.1 CORRELATION WITH OTHER BENCHMARKS

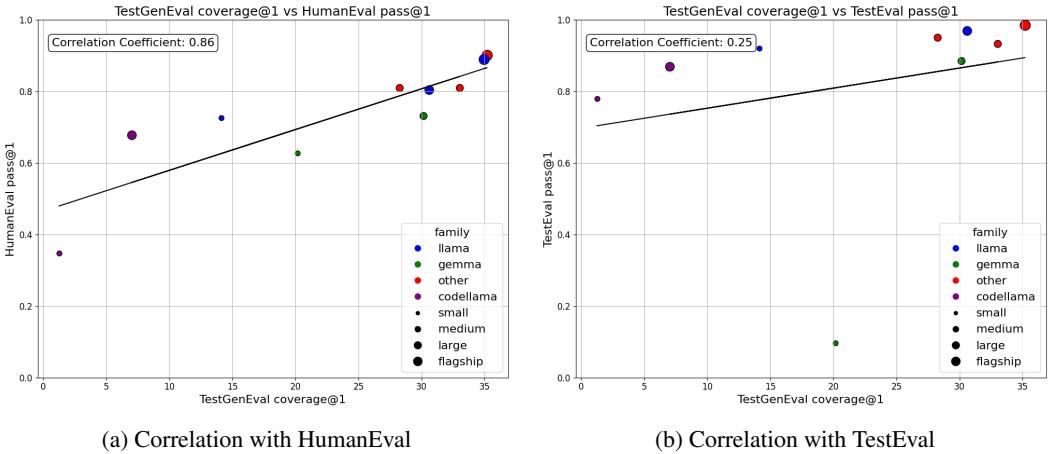

(a) Correlation with HumanEval     (b) Correlation with TestEval

Figure 6: Correlation with HumanEval (a SOTA code generation benchmark), and TestEval (a SOTA test generation benchmark). We find that there is a weak positive correlation between HumanEval and TESTGENEVAL, and with the exception of Gemma 9B there is a similar weak positive correlation between TestEval and TESTGENEVAL.

Figure 6a and Figure 6b display correlation between TESTGENEVAL, HumanEval (a code generation benchmark), and TestEval (a test generation benchmark). We find a weak positive correlation between TESTGENEVAL scores and other benchmarks. For HumanEval outliers include Gemma 9B (performs better at test generation than expected given code generation performance) and CodeLlama (performs worse at test generation than expected). For TestEval, the main outlier is Gemma 9B, with reasonable test generation performance on TESTGENEVAL, but very poor performance on TestEval (it fails to properly follow the prompt format for TestEval). Full details and a comparison against MMLU can be found in Appendix F.1.

### 4.1.2 EFFECT OF NUMBER OF SAMPLES

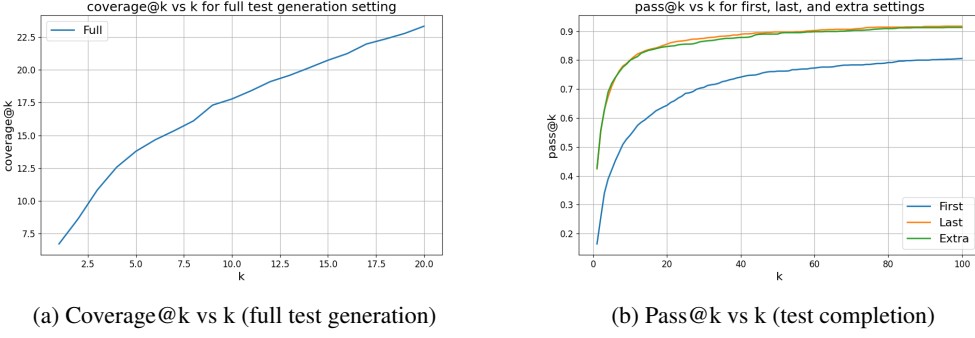

(a) Coverage@k vs k (full test generation)     (b) Pass@k vs k (test completion)

Figure 7: Effect of sampling more tests for all settings on Llama 3.1 8B. Coverage@k seems to gradually increase. Pass@k also increases more for lower k values and seems to plateau after k=20.

Figure 7a and Figure 7b show how performance in the full test generation and first, last and extra test completion settings changes as more tests are sampled for Llama 3.1 8B. For full test suite generation we sample 20 examples and measure coverage@k. For test completion we sample 100 examples and measure pass@k. For full test suite generation, we find that coverage seems to gradually increase, with no plateau in the first 20 generations. For test completion, we find that the first 5 samples improve performance the most, and performance gains seem to plateau after k=20 (the gain between k=20 and k=100 is minimal). Full details of this analysis can be found in Appendix F.3.

### 4.1.3 EFFECT OF CONTEXT WINDOW ON TESTGENEVAL PERFORMANCE

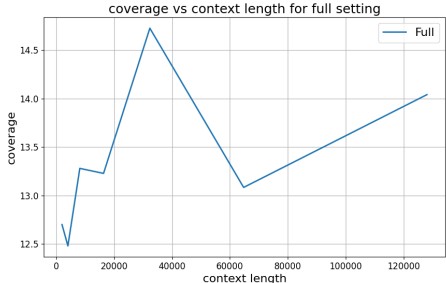

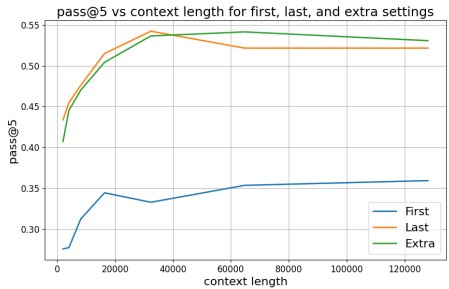

(a) Coverage by context length (full test generation)          (b) Pass@5 by context length (test completion)

Figure 8: Effect of context window for both our full setting (coverage of generated test suite) and for our test completion setting (pass@5 for our first, last and extra test completion settings). We find that context length generally helps test completion, however for test generation, even receiving parts of the file under test (measured by a lower context window) seems to be effective.

Figure 8a and Figure 8b show the effect of context length on both our coverage (full test generation) and on pass@5 (test completion). For test generation, we find that coverage only slightly improves with additional context; even seeing part of the code under test is enough for the model to generate a test suite (can mock inputs to various methods in the partial file). However for test completion context is more important, with pass@5 increasing with more context up until around 32k tokens where benefits decay. To complete tests, one must understand existing tests and their relationship with the code under test. Having the entire file helps contextualize what existing tests are testing, improving the performance of completed tests. Full details of this analysis can be found in Appendix F.4.

### 4.2 QUALITATIVE ANALYSIS

We highlight a case where TESTGENEVAL discriminates between our top three models (GPT-4o, Llama 405B and Codestral 22B). Additional examples comparing these three models can be found in Appendix F.6. Additionally, qualitative examples of common error types can be found in Appendix F.7 and examples of frontier problems (problems solved by at most one model and no models) in Appendix F.8.

```python
class QuerySet(AltersData):
    def __init__(self, model=None, query=None, ...):
        ...
    def repr(self):
        return "<%s %r>" % (self.__class__.__name__, data)
    ...
```

Listing 1: Query set class for managing data from database.

Our example involves a QuerySet class that manages records returned from a database. The class has no database dependencies. The initialization takes a model and query, which can also not be set if no database is being used. This is hard to test because it involves either mocking or creating a complex model object.

```python
def test_queryset_repr(self):
    queryset = QuerySet(model=None)
    assertEqual(repr(queryset), "<QuerySet []>")
```

Listing 2: Cut GPT-4o generated test

GPT-4o instantiates the QuerySet with no parameters (the simplest possible version of the test). It then tests all code methods, with all of the generated tests passing on the code under test.

```python
from django.db import connection, models

def test_query_set_init(self):
    model = models.Model()
    qs = QuerySet(model=model)
    self.assertEqual(qs.model, model)
```

Listing 3: Cut Llama 3.1 405B generated test

This is not the case with both Llama 3.1 405B and Codestral 22B. Both models ignore the empty case entirely and instead hallucinate invocations or imports of models in Django (instead these models should have mocked the model object and tested the null case). All models failing to properly mock the class under test's dependencies lead to low coverage of the source file, despite GPT-4o generating passing tests all when other models fail.

## 5 LIMITATIONS

We outline potential limitations with TESTGENEVAL. A more detailed set of limitations can be found in Appendix H.

**Overfitting to SWEBench repositories:** A potential limitation is that our benchmark is adapted from SWEBench and as a result risks models overfitting to this specific dataset. Currently, this does not seem to be a major issue, as model performance is low across the board. Even once models can achieve high coverage, there is the significantly harder task of achieving high mutation score (actually catching synthetic bugs introduced into the code under test). These multiple levels of difficulty and numerous tasks help mitigate the risk of a model overfitting to any one task specifically.

**Data contamination:** There is also a risk of data contamination in the pretraining data of models. To further understand data contamination, we measure perplexity of 10 randomly selected tests in TESTGENEVAL for Llama 3.1 8B and common frequent, and non recent code from GitHub with similar lengths. We find that the perplexity of this common GitHub code is lower than the 10 tests from TESTGENEVAL (1.6 vs 2.0), indicating that data is unlikely to be contaminated. This is further supported by the low performance of all models on TESTGENEVAL across the board.

**Compute cost of mutation score:** One other limitation is the compute cost of computing mutation score. Each synthetic bug we introduce to the code under test, requires an additional test suite execution. However, our results show that coverage and mutation score are highly correlated. Setting a timeout of one hour per mutation testing run, we only timeout on 20% of files, and get an average uncertainty of 1.1%. We provide an option to run TESTGENEVAL without mutation, enabling those who lack compute to still benefit from TESTGENEVAL.

## 6 RELATED WORK

**Test Generation Benchmarks:** Recently, there has been more effort to evaluate language models software testing capabilities, however these benchmarks typically still consist of small, self-contained projects. Common code generation benchmarks such as HumanEvalFix (Chen et al., 2021), MBPP Plus (Austin et al., 2021) and APPS (Hendrycks et al., 2021) can easily be adapted to test generation tasks. TestEval (Wang et al., 2024) measures test generation capabilities for Leet-Code problems of varying difficulties. Bhatia et al. (2024) measure test generation performance for modular code without external dependencies. While these benchmarks provide important execution metrics, the small size of these problems and solutions does not mirror realistic test generation.

There are also repository level test completion benchmarks, however these benchmarks often lack realistic execution metrics. TeCo (Nie et al., 2023) and ConTest (Villmow et al., 2021) provides a benchmark for completing unit tests, but only measures lexical metrics such as BLEU (Papineni et al., 2002) and ROUGE scores (Lin, 2004). ATLAS (Watson et al., 2020) and TOGA (Dinella et al., 2022) provide large scale benchmarks but for completing assertions rather than generating entire tests. SWT-Bench (Mündler et al., 2024) provides a benchmark for test method generation targeted as bug fixing PRs. Their task is also adjacent but measuring a specialized part of software testing (generating tests that fail on code prior to PR and pass after PR).

More recently, there has been efforts to create executable code benchmarks. SWEBench (Jimenez et al., 2024) measures the ability of language models to generate patches for failing PR tests. R2E (Jain et al., 2024) provides a collection of 400 repositories that can be executed, however they leverage equivalence test harnesses which look structurally different than human written unit tests. CruxEval (Gu et al., 2024) provides a large set of executable code snippets, however measures execution reasoning ability, which is a subset of the test generation task. TESTGENEVAL provides a large scale executable environment to measure test generation and test completion performance.

**Large Language Models for Code Generation:** Large language models (LLMs) have shown promising results on many software engineering tasks, including software testing (Rao et al., 2023; Black et al., 2022; Touvron et al., 2023a). Top open source models include Llama 3.1 (Dubey et al., 2024), DeepSeekCoder (DeepSeek-AI et al., 2024) and CodeStral [2]. Closed source models including GPT-4-o [3] and Claude 3.5 Sonnet [4] have also shown state of the art performance on a wide variety of code generation benchmarks. We measure performance for top open source models and GPT-4o, the current state of the art closed source model.

## 7 CONCLUSION

We introduce TESTGENEVAL, the first file-level benchmark for test generation and test completion. We release a lite version consisting of 160 code-test file pairs and a full benchmark comprising 1,210 file pairs from real open-source projects. Considering real-world settings, we employ coverage and mutation score to evaluate the models in our benchmark, as these metrics are closely related to the real-world quality of test suites. We perform a comprehensive evaluation of both open and closed source models on TESTGENEVAL. Additionally, we conduct a thorough error analysis, revealing that models struggle with reasoning about execution, frequently making assertion errors and failing to generate tests that run within the time limit. Overall, we believe TESTGENEVAL provides a complementary dataset to existing test generation datasets, offering a more challenging and larger-scale version of current benchmarks.

## 8 ETHICS STATEMENT

As models continue to grow in size and test generation capabilities improve, it is important to consider the broader impacts of test generation. While generally test generation is an important task for ensuring high quality software, generated tests can be used to assert that buggy code is correct (as seen by the oracle problem). All generated tests should therefore be checked and approved by developers. Furthermore, automated test generation techniques have the potential to automate quality assurance jobs, which can also potentially lead to negative societal impacts. We recommend practitioners be cognizant of the impacts of test generation work when building and deploying new techniques.

## 9 REPRODUCIBILITY STATEMENT

We took extensive efforts to ensure TESTGENEVAL is reproducible. We release code to run and extend TESTGENEVAL along with all docker images for each project in SWEBench. We release a website to view model generated tests for qualitative analysis. Our code and website is available

---

[2]https://mistral.ai/news/codestral/

[3]https://openai.com/index/hello-gpt-4o/

[4]https://www.anthropic.com/news/claude-3-5-sonnet

at `https://figshare.com/s/51171ae97cd21d233d4f`. We designed TESTGENEVAL's code to be easy to extend; all projects have individual docker images to run them, enabling those looking to extend TESTGENEVAL to simply create a new docker container and hook into our existing code.

## 10 ACKNOWLEDGMENTS

The authors would like to thank Ori Yoran and Pierre Chambon for their feedback on the paper, as well as Claire Le Goues for her support and mentorship. We also thank the CodeGen team for their extensive support and feedback throughout the project. Last but not least, we give a special thanks to Mei 🐕, an outstanding canine software engineering researcher, for providing support and motivation throughout this paper.

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

# Appendix

## Table of Contents

# A  BENCHMARK STATISTICS

This section contains more details on benchmark statistics. Both our TESTGENEVAL and TEST-GENEVALLITE splits consist of 11 repositories shown in Table 4.

Table 4: Benchmark repository information and datapoint distribution.

| Repository | URL | # Stars | # Lite | # Full |
|---|---|---|---|---|
| astropy/astropy | https://github.com/astropy/astropy | 4325 | 3 | 45 |
| django/django | https://github.com/django/django | 78287 | 66 | 451 |
| matplotlib/matplotlib | https://github.com/matplotlib/matplotlib | 19756 | 7 | 72 |
| mwaskom/seaborn | https://github.com/mwaskom/seaborn | 12264 | 2 | 12 |
| pallets/flask | https://github.com/pallets/flask | 67203 | 1 | 2 |
| pydata/xarray | https://github.com/pydata/xarray | 3523 | 3 | 45 |
| pylint-dev/pylint | https://github.com/pylint-dev/pylint | 5206 | 2 | 17 |
| pytest-dev/pytest | https://github.com/pytest-dev/pytest | 11719 | 12 | 67 |
| scikit-learn/scikit-learn | https://github.com/scikit-learn/scikit-learn | 59057 | 20 | 192 |
| sphinx-doc/sphinx | https://github.com/sphinx-doc/sphinx | 6264 | 2 | 70 |
| sympy/sympy | https://github.com/sympy/sympy | 12657 | 42 | 239 |

## A.1  TESTGENEVAL

We report detailed statistics for the TESTGENEVAL. These include the coverage of various files, unique generations, correlations between different settings and MMLU.

Figure 9a, Figure 9b and Figure 9c show the distribution of code lengths, test lengths, and repositories across TESTGENEVAL. Tests and code files are thousands of tokens long, as the benchmark is large in size. The repository distribution mirrors that of SWEBench and the TESTGENEVALLITE split repo distribution is also similar.

Figure 10a, and Figure 10b show the distribution of code methods and test methods. Generally, both code and tests methods are quite extensive across file pairs (average of 57 test methods and 58 code methods per file pair).

Figure 11 provides violin plots of the coverage data of gold tests in TESTGENEVAL. Coverage is generally high across all settings (first test alone adds on average 40% coverage) and by the end of the test suite, file level coverage is approximately 80%.

## A.2  TESTGENEVALLITE

We perform the same analysis on the TESTGENEVALLITE split of our benchmark. Many of the distributions and trends observed in TESTGENEVAL hold here too.

Figure 12a, Figure 12b, and Figure 12c show the distribution of code lengths, test lengths, and repositories in the TESTGENEVALLITE split. Both the distribution of repositories and lengths of code and test files approximately mirror those present in full TESTGENEVAL. The long lengths even in the TESTGENEVALLITE split require models that support long context lengths.

Figure 10a, and Figure 10b show the distribution of code methods and test methods for TEST-GENEVALLITE. Results mirror those of TESTGENEVAL; both code and test files in this file have many methods (average of 58 test methods and 50 code methods for each file pair).

Figure 14 provides violin plots of the coverage data of gold tests in the TESTGENEVALLITE split. TESTGENEVALLITE coverage distributions mirror those of TESTGENEVAL.

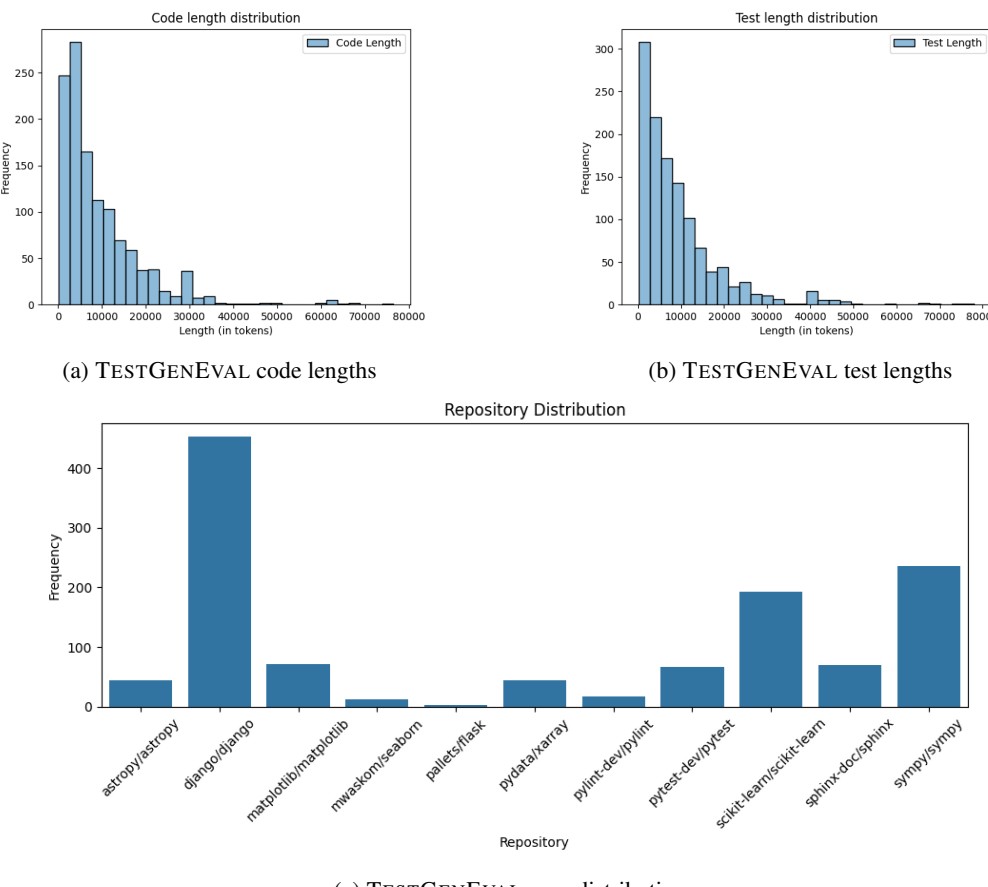

(a) TESTGENEVAL code lengths

(b) TESTGENEVAL test lengths

(c) TESTGENEVAL repo distribution

Figure 9: Code and test lengths across TESTGENEVAL. Both the code and test files in TEST-GENEVAL are from real world projects and thus are often 10,000+ tokens long. Also shows the distribution of repositories in TESTGENEVAL.

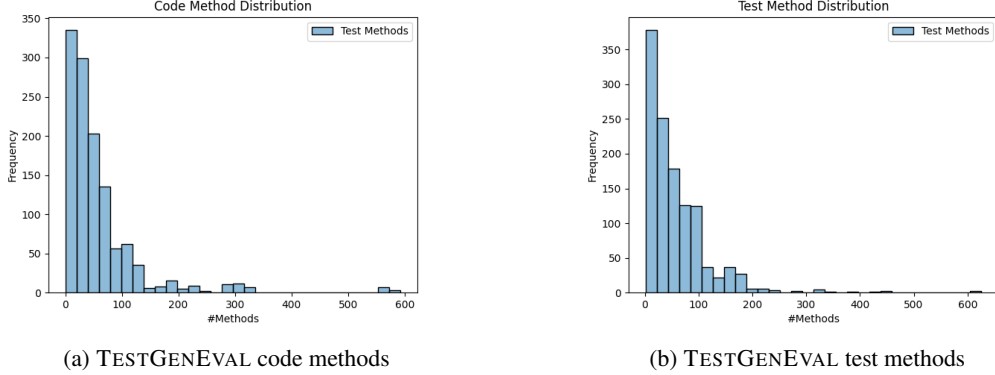

(a) TESTGENEVAL code methods

(b) TESTGENEVAL test methods

Figure 10: Distribution of code and test methods for TESTGENEVAL.

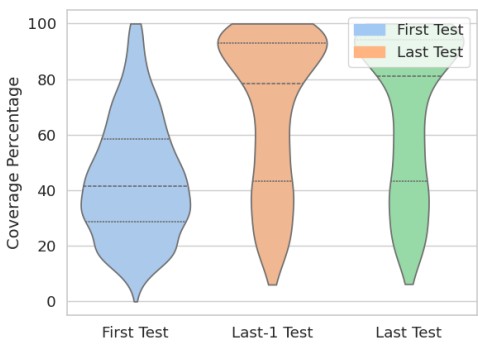

Figure 11: TESTGENEVAL gold test coverage

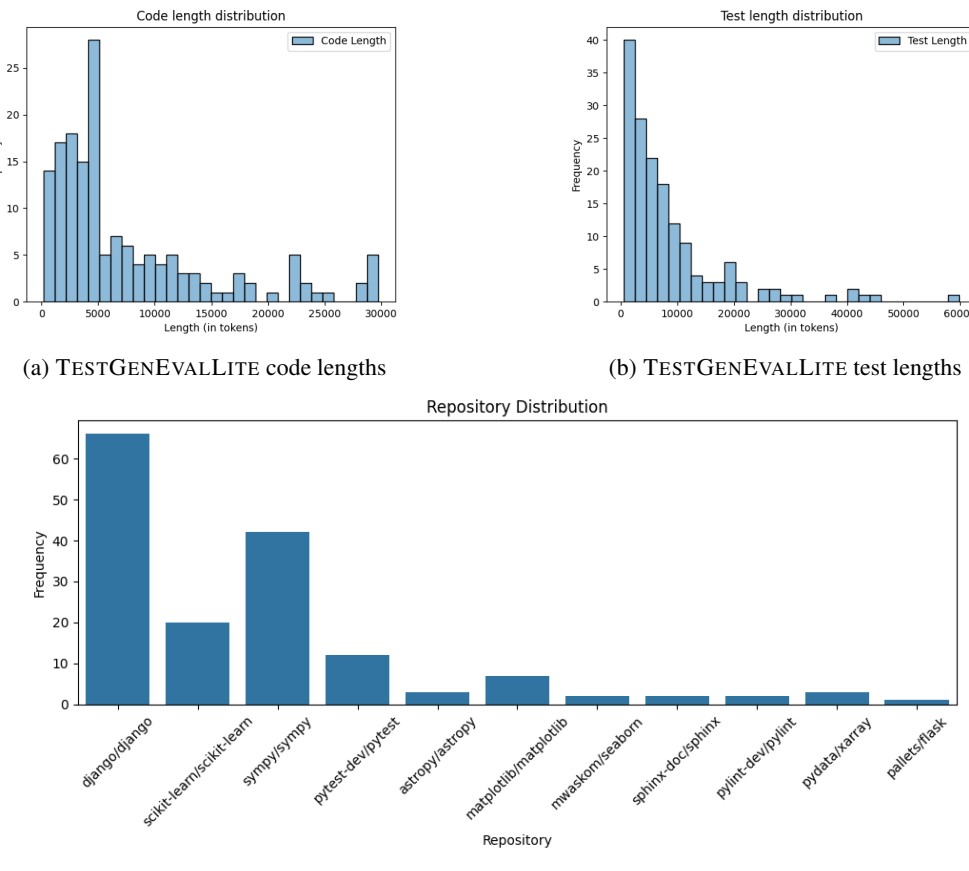

(a) TESTGENEVALLITE code lengths

(b) TESTGENEVALLITE test lengths

(c) TESTGENEVALLITE repo distribution

Figure 12: Distributions in the TESTGENEVALLITE dataset

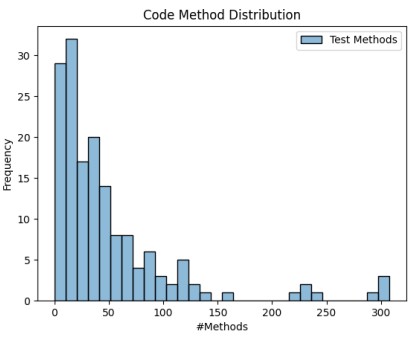

(a) TESTGENEVALLITE code methods

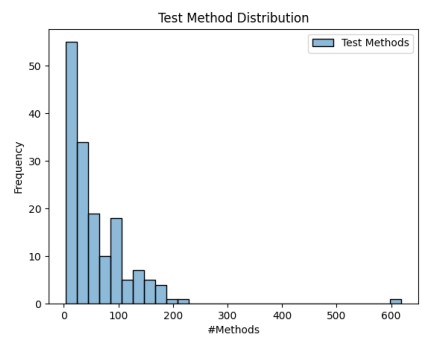

(b) TESTGENEVALLITE test methods

Figure 13: Distribution of code and test methods for TESTGENEVAL.

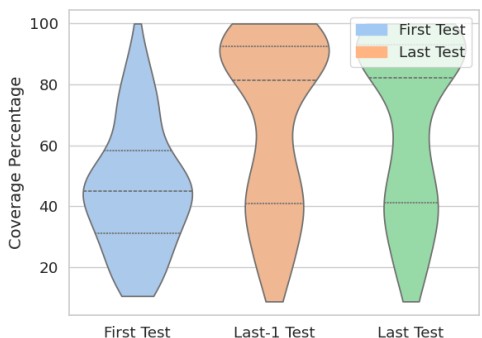

Figure 14: TESTGENEVALLITE gold test coverage

## B  MODEL URLS

We used GPT-4o on 08/29/2024. Table 5 shows the model name and model URL for all open source models on HuggingFace.

Table 5: Model Names and URLs

| Model Name | Model URL |
|---|---|
| CodeLlama (7B) | https://huggingface.co/meta-llama/CodeLlama-7b-Instruct-hf |
| CodeLlama (70B) | https://huggingface.co/meta-llama/CodeLlama-70b-Instruct-hf |
| Llama 3.1 (8B) | https://huggingface.co/meta-llama/Meta-Llama-3.1-8B-Instruct |
| Llama 3.1 (70B) | https://huggingface.co/meta-llama/Meta-Llama-3.1-70B-Instruct |
| Llama 3.1 (405B) | https://huggingface.co/meta-llama/Meta-Llama-3.1-405B-Instruct |
| Codestral (22B) | https://huggingface.co/mistralai/Codestral-22B-v0.1 |
| DeepSeekCoder-Lite (16B) | https://huggingface.co/deepseek-ai/DeepSeek-Coder-V2-Lite-Instruct |
| Gemma-2 (9B) | https://huggingface.co/google/gemma-2-9b-it |
| Gemma-2 (27B) | https://huggingface.co/google/gemma-2-27b-it |

## C  PROMPTS

For each setting we prompt with different contexts. We provide the code under test as part of the prompt for all settings. We also provide the preamble of the test file (imports and test setup) for first test completion, the entire test file minus the last test for last test completion and the entire test file for extra test completion. For our full file test generation setting, we only prompt with the code under test. Since we use a large variety of models, our prompts are slightly different for each class of model.

---

**System Prompt**

```
You are an expert Python automated testing assistant.  Your
job is to generate a test file given a code file.
```

---

**Instructions**

```
Below is a code file:
```python
{code_src}
```

The code file is called:  {code_filename}

Your job is to output a corresponding unit test file that
obtains high coverage and invokes the code under test.

Here are some examples of how to import {code_filename}, (you
should use these as reference)

```python
{imports}
```

Each unit test must be a function starting with test_.  Include
all your test imports and setup before your first test.  Do
not run the tests in the file, just output a series of tests.
Do not include a main method to run the tests.

Only output the unit test Python file in this format:

```python
Unit test Python code (file level)
```
```

Figure 15: Prompt for the full test suite generation setting. For models that use specific format for chat, we format this appropriately to chat format.

**System Prompt**

```
You are an expert Python software testing assistant.  Your
job is to complete the next test given a code file an some
existing test context.
```

**Instructions**

```
Below is a code file:
```python
{code_src}
```

And the current unit test file
```python
{test_src}
```

Your job write the Python code the next test in the file.
Ideally your next test should improve coverage of the existing
unit test file for the code file.

Only output the next unit test, preserve indentation and
formatting.  Do not output anything else.  Format like this:

```python
Next unit test Python code
```
```

Figure 16: Prompt for the test completion settings (first, last, extra). For models that use specific format for chat, we format this appropriately to chat format.

## D   EXPERIMENT SETUP

We outline the metrics used for evaluation for each of our tasks.

### D.1   MODELS

We evaluate a selection of models on TESTGENEVAL to better understand how models of different sizes and families perform at test suite generation and test completion. We include a mixture of small (less than 9B params), medium (between 9B and 27B params), large (approximately 70B params) and flagship (greater than 70B params) models. This includes open source models (CodeLlama, Llama 3, DeepSeekCoder 2, Codestral, and Gemma 2) of varying sizes and GPT-4o, a state of the art closed source model.

We choose the Llama and Gemma families of models to understand the effects of size on model performance in conjunction with their high scores on code benchmarks such as HumanEval. We also include DeepSeekCoder, CodeLlama and Codestral due to their code specialization. Finally, we include GPT-4o due to its state of the art performance on numerous code generation benchmarks.

### D.2   METRICS

We report metrics for both our test generation (Appendix D.2.1) and test completion (Appendix D.2.2) tasks.

#### D.2.1   TEST GENERATION METRICS

**All pass@1:** All pass@1 measures if the generated test suite passes when run on the code under test. This penalizes more verbose models, as the likelihood of an error increases with each additional test generated in a test suite.

**Any pass@1:** Any pass@1 measures if any test in the generated test suite passes when run on the code under test. We include this to not penalize models that generate longer test suites.

**Coverage:** Coverage measures the proportion of lines in the file under test executed by the test suite. Ideally, a high quality test suite should execute a large percentage of the lines in the code under test.

**Coverage@pass:** Coverage@pass measures the coverage of only the passing tests. Models with high coverage@pass may not successfully generate tests for many of the problems in TEST-GENEVAL, but for the ones they do, they obtain high coverage.

**Mutation score:** The main limitation of coverage and pass@1 is that they can be potentially gamed (a model can invoke all functions in the file under test without testing anything and achieve 100% coverage and 100% pass@1). To compute mutation score we inject synthetic bugs into the code under test. We then measure the percentage of bugs detected by the test suite (should pass on the original code and fail on the buggy code). Unlike other metrics, mutation score is much harder to game, however it is compute costly, as we have to execute the entire test suite for each bug. We rank by coverage, as models still have relatively low coverage across the board and it is more compute efficient for the community to run (we allow for TESTGENEVAL to be run omitting mutation score).

We use cosmic-ray[5] to generate mutants and use the default set of mutation operators[6]. This default set of operators follows best practices defined by the mutation testing community (Derezinska & Halas, 2014; Offutt et al., 1996). Cosmic ray has 565 stars and commonly in mutation testing research (Sánchez et al., 2022; Deb et al., 2024). We choose to not filter out mutation operators to achieve the most granular results possible (with no filtering there is only a 1.06% uncertainty in mutation score results).

**Mutation score@pass:** Mutation score@pass measures the the mutation score of only passing tests. Mutation score@pass is similar to coverage@pass, measuring how the quality of generated tests, rather than their coverage of all problems in TESTGENEVAL.

---

[5] https://github.com/sixty-north/cosmic-ray
[6] https://github.com/sixty-north/cosmic-ray/tree/master/src/cosmic_ray/operators

### D.2.2 TEST COMPLETION METRICS

**Pass@k:** Pass@k measures if any of k test generated pass when added to the existing test suite for the code under test. We rank by this metric, as coverage improvement is near 0 for 2/3 settings for TESTGENEVAL.

**Avg pass@k:** Average pass@k measures the average proportion of k generated tests that pass. This provides an idea of how many erroneous tests a model will generate before generating a correct test.

**Coverage improvement:** Coverage improvement measures the change in line coverage when adding the generated test. Ideally, newly added tests should improve overall code coverage. We choose not to report mutation score improvement here, due to computational cost and already near 0 coverage improvement of generated tests.

**Coverage improvement@pass:** Coverage improvement@pass measures the change in line coverage for only problems where generated tests pass the test suite. High coverage improvement@pass indicates that a model may be good at generating high coverage test completion, but could still struggle with generating passing test completions for all problems.

# E  RESULTS

We report our full set of results on both TESTGENEVAL and TESTGENEVALLITE (results across both splits are similar). We also report complete test completion results for the extra setting and add the average pass@k metric.

## E.1  TESTGENEVAL

We report performance on both the test generation and test completion tasks for TESTGENEVAL.

### E.1.1  TEST GENERATION

Table 6: Full test suite generation. All results shown for temperature=0.2. We report all pass@1, any pass@1, coverage, coverage of only passing tests, mutation score and mutation score of only passing tests.

| Model | All@1 | Any@1 | Cov | Cov@Pass | Mut | Mut@Pass |
|---|---|---|---|---|---|---|
| **Small Models** | | | | | | |
| CodeLlama 7B | 3.2% | 4.1% | 1.2% | 30.3% | 0.5% | 11.0% |
| Gemma 9B | 3.5% | 42.1% | 20.2% | 47.9% | 9.0% | 21.5% |
| Llama 3.1 8B | 3.1% | 30.5% | 14.1% | 46.3% | 6.8% | 22.4% |
| **Medium Models** | | | | | | |
| DeepSeekCoder 16B | 23.0% | 63.7% | 28.2% | 44.3% | 12.1% | 19.1% |
| Gemma 27B | 7.4% | 57.7% | 30.1% | 52.3% | 14.6% | 25.5% |
| Codestral 22B | **26.8%** | 72.7% | 33.0% | 45.4% | 14.2% | 19.7% |
| **Large Models** | | | | | | |
| CodeLlama 70B | 14.0% | 22.7% | 7.0% | 32.4% | 2.5% | 11.4% |
| Llama 3.1 70B | 7.8% | 60.7% | 30.6% | 50.4% | 15.4% | 25.6% |
| **Flagship Models** | | | | | | |
| GPT-4o | 7.5% | 64.0% | **35.2%** | **54.9%** | **18.8%** | **29.4%** |
| Llama 3.1 405B | 17.7% | **73.1%** | 35.0% | 47.9% | 16.4% | 22.5% |

Table 6 shows full test suite performance for TESTGENEVAL. Smaller models struggle to generate any tests that pass on complex code files present in TESTGENEVAL. Codestral 22B can generate a passing test in 72.7% of all problems, however still has lower coverage and mutation score than GPT-4o. This indicates despite generating tests for more problems, the tests generated by Codestral 22B tend to be lower quality than those generated by GPT-4 (indicated by a approximately 10% lower coverage of passing tests).

### E.1.2  TEST COMPLETION

Table 7 shows the pass rates across first, last, and extra test completion settings. Extra test completion setting results mirror those of last test completion, with Codestral 22B outperforming existing models.

Table 8 shows the average pass@5 rates along with coverage improvements for all settings. Coverage improvement for last and extra settings is near 0; it is hard to improve coverage for a near complete test suite. Average pass@5 is interestingly lower for GPT-4o, despite higher coverage, meaning that it would take more attempts to generate a passing test completion, but the quality of the completion is better.

Figure 17a, Figure 17b, Figure 17c show the number of unique generations for each model across temperatures 0.2 and 0.8. Of all models, CodeLlama models are the least diverse, with not a big difference between 0.2 and 0.8 temperature, while Llama 3.1 generates far more unique solutions.

Table 7: Pass rates for different models in first, last, and extra completion settings. Pass@1 is for temperature=0.2, Pass@5 is for temperature=0.8.

| Model | First | | Last | | Extra | |
|---|---|---|---|---|---|---|
| | Pass@1 | Pass@5 | Pass@1 | Pass@5 | Pass@1 | Pass@5 |
| **Small Models** | | | | | | |
| CodeLlama 7B | 4.2% | 19.8% | 6.9% | 24.0% | 5.2% | 22.9% |
| Gemma 9B | 8.4% | 18.0% | 21.4% | 46.4% | 18.9% | 46.7% |
| Llama 3.1 8B | 14.4% | 33.3% | 32.0% | 54.3% | 31.8% | 53.8% |
| **Medium Models** | | | | | | |
| DeepSeekCoder 16B | 18.6% | 47.0% | 17.0% | 62.8% | 15.5% | 61.7% |
| Gemma 27B | 12.7% | 33.7% | 32.2% | 62.2% | 31.7% | 66.8% |
| Codestral 22B | **38.3%** | 61.7% | **50.4%** | **74.3%** | **48.3%** | **75.7%** |
| **Large Models** | | | | | | |
| CodeLlama 70B | 0.5% | 30.2% | 0.9% | 50.7% | 0.5% | 52.8% |
| Llama 3.1 70B | 19.3% | 46.4% | 35.0% | 61.9% | 36.4% | 61.2% |
| **Flagship Models** | | | | | | |
| GPT-4o | 31.9% | **63.5%** | 32.6% | 66.6% | 30.4% | 63.5% |
| Llama 3.1 405B | 32.1% | 57.7% | 42.6% | 72.3% | 42.4% | 74.5% |

Table 8: Coverage improvements, including coverage improvements for passing tests, and average pass@5 for first, last, and extra completion settings. Models struggle to improve coverage, and in general report substantially worse average pass@5 than pass@5.

| Model | First | | | Last | | | Extra | | |
|---|---|---|---|---|---|---|---|---|---|
| | +Cov | +Cov@P | Avg@5 | +Cov | +Cov@P | Avg@5 | +Cov | +Cov@P | Avg@5 |
| **Small Models** | | | | | | | | | |
| CodeLlama 7B | 6.7% | 33.9% | 4.7% | 0.0% | 0.1% | 5.7% | 0.0% | 0.1% | 5.3% |
| Gemma 9B | 6.7% | 37.3% | 8.2% | 0.1% | 0.2% | 17.4% | 0.1% | 0.2% | 16.5% |
| Llama 3.1 8B | 12.8% | 38.5% | 10.5% | 0.1% | 0.2% | 23.1% | 0.0% | 0.1% | 23.5% |
| **Medium Models** | | | | | | | | | |
| DeepSeekCoder 16B | 19.0% | 40.4% | 16.4% | 0.2% | 0.3% | 22.2% | 0.1% | 0.1% | 21.9% |
| Gemma 27B | 13.6% | 40.2% | 13.2% | 0.1% | 0.2% | 28.0% | 0.1% | 0.1% | 29.0% |
| Codestral 22B | 24.0% | 39.1% | 31.2% | 0.4% | 0.5% | **43.2%** | 0.2% | 0.2% | **43.5%** |
| **Large Models** | | | | | | | | | |
| CodeLlama 70B | 11.2% | 37.5% | 7.5% | 0.0% | 0.0% | 14.2% | 0.0% | 0.0% | 15.4% |
| Llama 3.1 70B | 18.7% | 40.5% | 18.4% | **0.5%** | **0.7%** | 29.5% | 0.1% | 0.2% | 29.0% |
| **Flagship Models** | | | | | | | | | |
| GPT-4o | **26.9%** | **42.4%** | **32.4%** | **0.5%** | **0.7%** | 31.5% | **0.2%** | **0.3%** | 28.8% |
| Llama 3.1 405B | 21.6% | 37.5% | 31.3% | 0.3% | 0.4% | 40.9% | 0.1% | 0.2% | 41.4% |

(a) First setting      (b) Last setting      (c) Extra setting

Figure 17: Unique generations by model and temperature for first, last and extra test completion settings (out of maximum of 5 generations).

## E.2 TESTGENEVALLITE

We also report our full set of results on our TESTGENEVALLITE split. We find that results generally mirror those of TESTGENEVAL.

### E.2.1 TEST GENERATION

| Model | All@1 | Pass@1 | Cov | Cov@Pass | Mut | Mut@Pass |
|---|---|---|---|---|---|---|
| | | | **Small Models** | | | |
| CodeLlama 7B | 3.8% | 4.4% | 2.1% | 47.5% | 0.8% | 18.3% |
| Gemma 9B | 7.5% | 45.0% | 19.6% | 43.6% | 8.1% | 17.9% |
| Llama 3.1 8B | 5.0% | 29.4% | 12.1% | 41.3% | 5.1% | 17.9% |
| | | | **Medium Models** | | | |
| DeepSeekCoder 16B | 24.4% | 64.4% | 27.3% | 42.5% | 11.6% | 18.0% |
| Gemma 27B | 11.9% | 58.8% | 28.4% | 48.3% | 12.8% | 22.1% |
| Codestral 22B | **30.0%** | 71.9% | 31.7% | 44.1% | 12.3% | 17.3% |
| | | | **Large Models** | | | |
| CodeLlama 70B | 18.8% | 25.6% | 8.0% | 32.8% | 1.8% | 7.7% |
| Llama 3.1 70B | 10.0% | 60.6% | 30.3% | 50.0% | **15.1%** | 25.4% |
| | | | **Flagship Models** | | | |
| GPT-4o | 5.0% | 55.0% | 29.7% | **54.0%** | 14.7% | **27.3%** |
| Llama 3.1 405B | 18.1% | **74.4%** | **34.1%** | 45.9% | 14.7% | 19.9% |

Table 9: Full test suite generation. All results shown for temperature=0.2

Table 9 shows pass@1, all pass@1 coverage and mutation score for small, medium and large models. Similar to TESTGENEVAL, smaller models perform significantly worse than their larger counterparts. All pass@1 penalizes how verbose a model is, thus less verbose models such as Codestral 22B have a much higher all pass@1 than verbose models such as GPT-4o, which generate many tests.

Coverage and mutation score remain low across all models. Similar to TESTGENEVAL, GPT-4o and Llama 3.1 405B performs highly. Mutation score is even lower than coverage, implying that models are not able to catch all induced bugs.

### E.2.2 TEST COMPLETION

Table 10 shows pass@1, and pass@5 for all settings. Similar to TESTGENEVAL split, model performance generally increases as more of the test file is provided as context (extra pass@5 is higher than both last pass@5 and first pass@5). Table 11 shows the average pass@5 and coverage improvement across all three settings. Similar to TESTGENEVAL, models struggle to add coverage to an existing test suite, performing best at first test generation, where there is no coverage (or minimal coverage from test setup) and thus it is easiest to add new coverage. For the last and extra test generation testing, the model generates nearly 0 new coverage, testing already tested functions, rather than attempting to test new functions. This represents a frontier of test generation (improving coverage of an already high coverage test suite). Similar to TESTGENEVAL, Codestral has a higher average pass@5 than GPT-4o, despite GPT-4o generating tests with higher coverage.

Figure 18a, Figure 18b, Figure 18c show the number of unique generations for each model across temperatures 0.2 and 0.8. Similar to the full TESTGENEVAL, CodeLlama generates the fewest unique solutions, while models such as Llama 3.1 8B generate a large number of unique solutions.

Table 10: Pass rates for different models in first, last, and extra completion settings. Pass@1 is for temperature=0.2, Pass@5 is for temperature=0.8.

| | First | | Last | | Extra | |
|---|---|---|---|---|---|---|
| **Model** | **Pass@1** | **Pass@5** | **Pass@1** | **Pass@5** | **Pass@1** | **Pass@5** |
| **Small Models** | | | | | | |
| CodeLlama 7B | 6.9% | 25.0% | 7.5% | 22.5% | 6.9% | 21.9% |
| Gemma 9B | 6.9% | 18.1% | 18.8% | 45.0% | 25.0% | 45.6% |
| Llama 3.1 8B | 14.4% | 41.2% | 35.0% | 56.2% | 31.9% | 51.2% |
| **Medium Models** | | | | | | |
| DeepSeekCoder 16B | 18.8% | 48.8% | 20.6% | 60.0% | 13.1% | 60.6% |
| Gemma 27B | 16.2% | 35.6% | 31.9% | 63.7% | 30.6% | 63.1% |
| Codestral 22B | **43.1%** | **67.3%** | **51.2%** | **75.0%** | **47.5%** | **75.0%** |
| **Large Models** | | | | | | |
| CodeLlama 70B | 1.2% | 26.9% | 0.0% | 61.9% | 0.0% | 56.9% |
| Llama 3.1 70B | 21.9% | 48.1% | 36.2% | 59.4% | 41.2% | 64.4% |
| **Flagship Models** | | | | | | |
| GPT-4o | 36.2% | 66.9% | 26.9% | 60.6% | 26.2% | 61.3% |
| Llama 3.1 405B | 33.8% | 60.0% | 38.8% | 68.1% | 40.6% | 71.2% |

Table 11: Coverage improvements, including coverage improvements for passing tests, and average pass@5 for first, last, and extra completion settings. Models struggle to improve coverage, and in general report substantially worse average pass@5 than pass@5.

| | First | | | Last | | | Extra | | |
|---|---|---|---|---|---|---|---|---|---|
| **Model** | **+Cov** | **+Cov@P** | **Avg@5** | **+Cov** | **+Cov@P** | **Avg@5** | **+Cov** | **+Cov@P** | **Avg@5** |
| **Small Models** | | | | | | | | | |
| CodeLlama 7B | 9.3% | 37.3% | 5.8% | 0.0% | 0.1% | 5.0% | 0.0% | 0.0% | 5.3% |
| Gemma 9B | 7.9% | **43.7%** | 6.8% | 0.1% | 0.2% | 17.1% | 0.1% | 0.1% | 16.1% |
| Llama 3.1 8B | 16.3% | 39.5% | 10.7% | 0.0% | 0.0% | 26.4% | 0.1% | 0.1% | 23.8% |
| **Medium Models** | | | | | | | | | |
| DeepSeekCoder 16B | 20.6% | 42.3% | 17.9% | 0.1% | 0.1% | 21.5% | 0.1% | 0.1% | 20.1% |
| Gemma 27B | 15.5% | 43.6% | 14.5% | 0.1% | 0.1% | 29.1% | 0.0% | 0.0% | 27.5% |
| Codestral 22B | 26.4% | 39.4% | 32.5% | 0.2% | 0.3% | **48.0%** | 0.1% | 0.1% | 43.8% |
| **Large Models** | | | | | | | | | |
| CodeLlama 70B | 8.7% | 33.3% | 7.0% | 0.0% | 0.0% | 17.0% | 0.0% | 0.0% | 15.7% |
| Llama 3.1 70B | 18.8% | 39.6% | 16.9% | 0.1% | 0.2% | 29.1% | 0.1% | 0.1% | 32.5% |
| **Flagship Models** | | | | | | | | | |
| GPT-4o | **28.2%** | 42.2% | **34.4%** | **0.3%** | **0.6%** | 28.8% | **0.3%** | **0.5%** | 28.1% |
| Llama 3.1 405B | 20.9% | 35.2% | 33.4% | 0.2% | 0.4% | 41.3% | 0.1% | 0.2% | **43.9%** |

(a) First setting      (b) Last setting      (c) Extra setting

Figure 18: Unique generations by model and temperature for first, last and extra test completion settings (out of maximum of 5 generations).

# F ANALYSIS

## F.1 CORRELATION WITH OTHER BENCHMARKS

### F.1.1 TESTGENEVAL

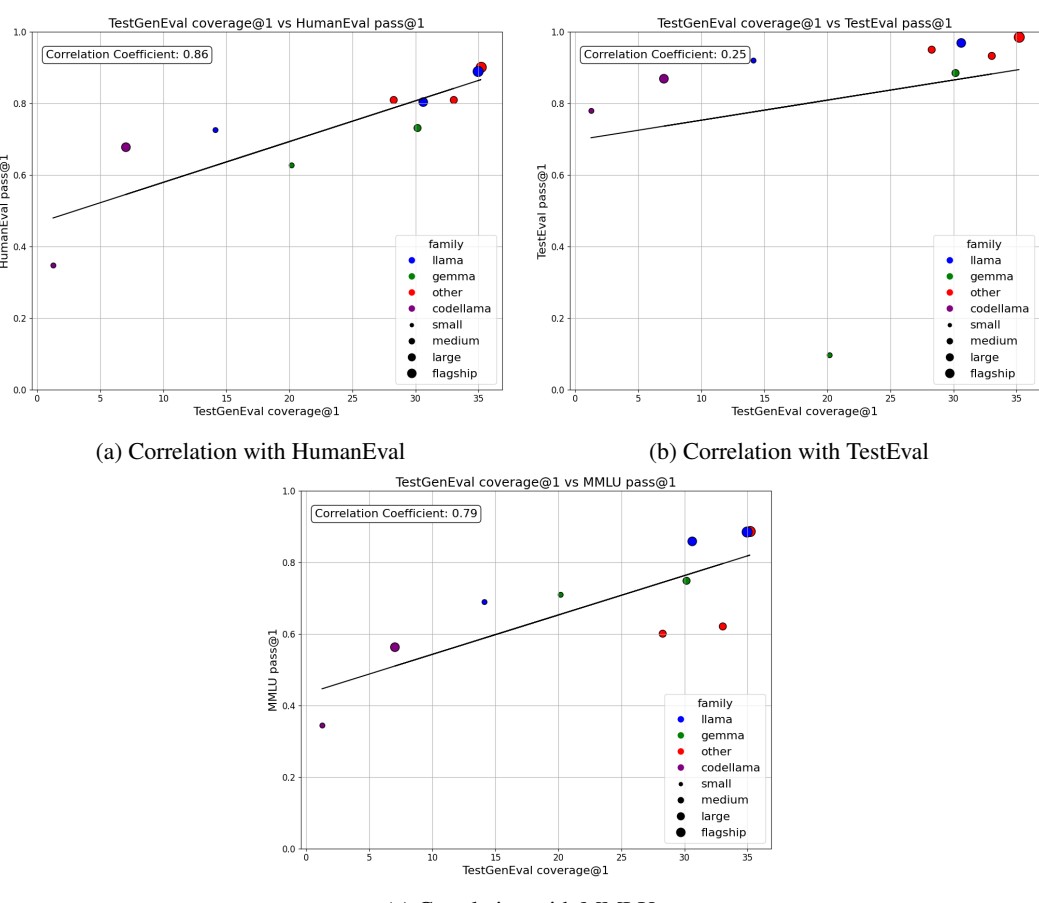

(a) Correlation with HumanEval     (b) Correlation with TestEval

(c) Correlation with MMLU

Figure 19: Correlation with HumanEval (a SOTA code generation benchmark), TestEval (a SOTA test generation benchmark) and MMLU (a NLP benchmark). We find that there is a weak positive correlation between HumanEval and TESTGENEVAL, and with the exception of Gemma 9B there is a similar weak positive correlation between TestEval and TESTGENEVAL. MMLU correlation is weaker, with code generation performing much better on TESTGENEVAL than MMLU.

We perform correlation analysis between TESTGENEVAL in our full test generation setting and other benchmarks. We select two benchmarks for comparison: HumanEval (a widely used code generation benchmark) and TestEval (a existing test generation benchmark).

Figure 19a shows correlation between TESTGENEVAL and HumanEval. Correlation between HumanEval (a code generation task) and TESTGENEVAL is higher with a correlation coefficient of 0.86. The major outlier here is CodeLlama 7B with a very low TESTGENEVAL and HumanEval score. Another outlier is Gemma 9B, with a lower HumanEval score than expected given its TESTGENEVAL score. The moderate correlation illustrates that there is still information added by adding TESTGENEVAL, but that it does indeed correlate with code generation performance.

Figure 19b shows correlation between TESTGENEVAL and TestEval, a smaller scale test generation benchmark. Generally all models tend to perform, well on TestEval, however Gemma 9B performs significantly worse than expected. This is because the instruct model does not follow the prompt provided by TestEval authors, instead generating code snippets. Additionally, there does is minimal

differentiation between models, with some models performing highly on TestEval (Llama 3.1 70B and CodeLlama 70B), but significantly worse on TESTGENEVAL.

Figure 19c shows correlation between TESTGENEVAL and MMLU, a state of the art natural language processing benchmark. Code generation models such as Codestral and DeepSeekCoder perform significantly worse on MMLU than on TESTGENEVAL. Otherwise, the performance is generally correlated.

## F.1.2 TESTGENEVALLITE

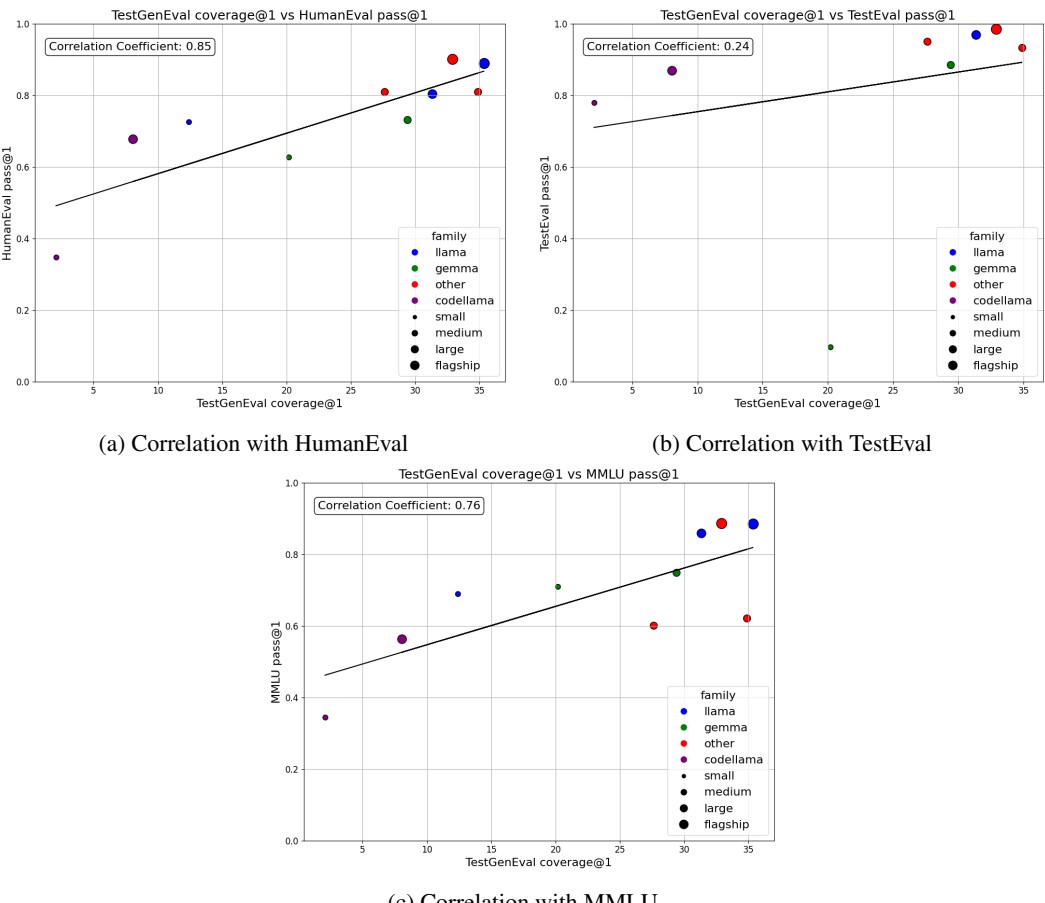

(a) Correlation with HumanEval

(b) Correlation with TestEval

(c) Correlation with MMLU

Figure 20: Correlation with HumanEval (a SOTA code generation benchmark), TestEval (a SOTA test generation benchmark) and MMLU (a NLP benchmark) with TESTGENEVALLITE. Correlations are similar to the full split of our benchmark.

Figure 20a, Figure 20c, and Figure 20b display the correlations with HumanEval, MMLU, and TestEval datasets, respectively, in the TESTGENEVALLITE dataset. Similar to TESTGENEVAL, we find that HumanEval has the highest correlation of any benchmark, however there are some outliers such as GPT-4o and Llama 3.1 8B having a higher HumanEval score. MMLU correlation is less strong, with code models such as Codestral and DeepSeekCoder scoring significantly lower on MMLU than on TESTGENEVAL. Finally, TestEval has the lowest correlation, with Gemma 9B failing to follow the TestEval prompt.

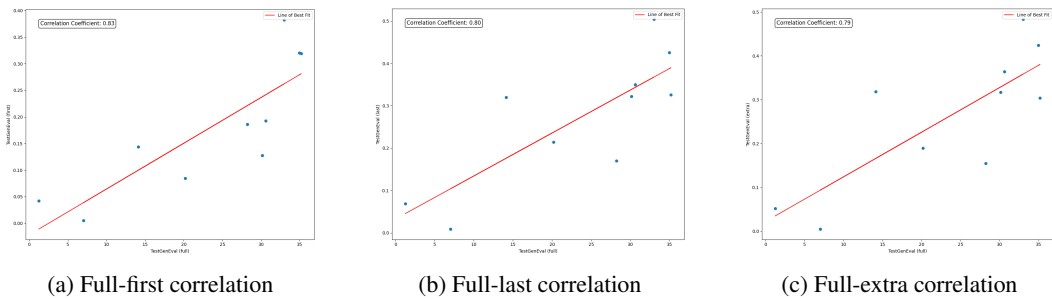

(a) Full-first correlation      (b) Full-last correlation      (c) Full-extra correlation

Figure 21: Correlations between the full dataset and the first, last, and extra subsets in the SWEBench dataset

## F.2 CORRELATION BETWEEN SETTINGS AND MODELS

### F.2.1 TESTGENEVAL

Figure 21a, Figure 21b, and Figure 21c show the correlations between the full dataset and the first, last, and extra subsets, respectively, in the TESTGENEVAL. Our full test generation setting has a relatively high correlation with other settings, however it is not a perfect correlation, with some models performing better on one setting than others.

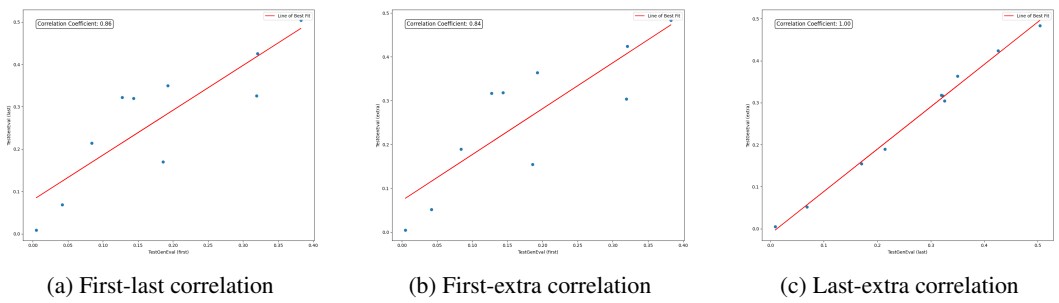

(a) First-last correlation      (b) First-extra correlation      (c) Last-extra correlation

Figure 22: Correlations between the full dataset and the first, last, and extra subsets in TEST-GENEVAL

Figure 22a, Figure 22b, and Figure 22c show the correlations between the test completion settings. First and last and first and extra test completion settings have a highly positive but not perfect correlation. Interestingly, last test generation and extra test generation have perfect correlation, indicating that models performing well in one setting will in the other setting.

Figure 23 shows the correlation of all models in their ability to solve the TESTGENEVAL full test generation setting. Models in the same families have the highest correlation (Gemma 9B and Gemma 27B for example, and Llama 3.1 8B and Llama 3.1 70B). Gemma 27B and Codestral 22B also are very correlated in their ability to solve problems in the full test suite generation setting.

Figure 24a, Figure 24b, and Figure 24c show the correlations of all models in their ability to solve first, last, and extra test completion tasks in TESTGENEVAL. Trends from the model predictions in the full test suite generation continue here, with models in the same family most correlated in their ability to solve tasks. Llama 3.1 405B and Codestral 22B also share some similarities in their first, last and extra test settings problems solved, as well.

### F.2.2 TESTGENEVALLITE

Figure 25a, Figure 25b, and Figure 25c show the correlations between the full test generation and the first, last, and extra completion settings, respectively, in the TESTGENEVALLITE split. Similar to TESTGENEVAL, correlation remains high between the full test suite generation and test completion settings. However, there are still cases where a model is better at one setting than others, indicating there is information gained by having all settings.

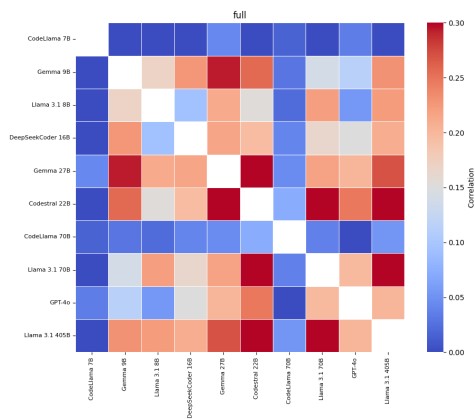

Figure 23: TESTGENEVAL full model correlation

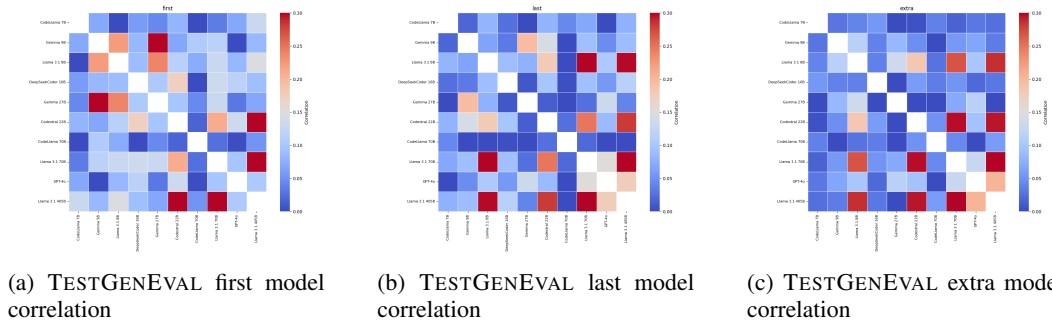

(a) TESTGENEVAL first model correlation

(b) TESTGENEVAL last model correlation

(c) TESTGENEVAL extra model correlation

Figure 24: Correlations for various test completion settings between all models in the TEST-GENEVAL.

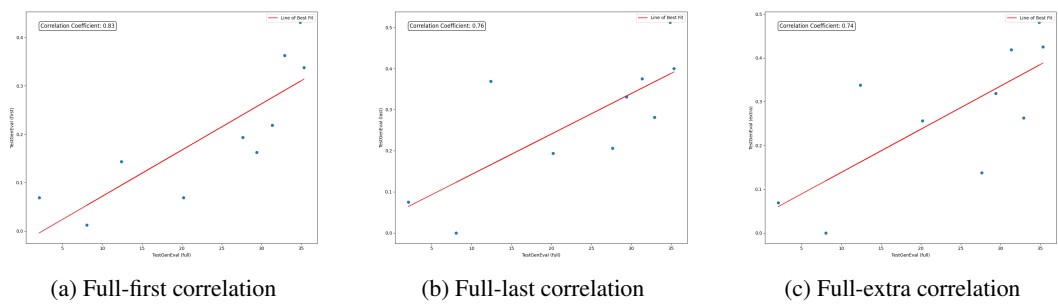

(a) Full-first correlation

(b) Full-last correlation

(c) Full-extra correlation

Figure 25: Correlations between the full test generation and the first, last, and extra completion settings in the TESTGENEVALLITE split

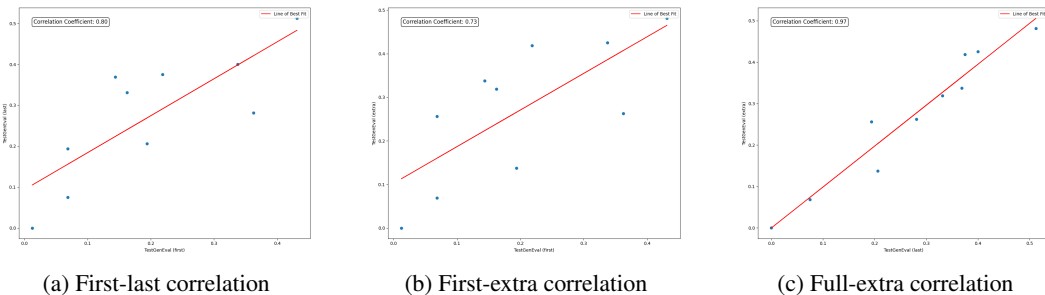

(a) First-last correlation

(b) First-extra correlation

(c) Full-extra correlation

Figure 26: Correlations between the full dataset and the first, last, and extra subsets in the TEST-GENEVALLITE split

Figure 26a, Figure 26b, and Figure 26c show the correlations between the test completion settings. The first test completion setting is highly correlated with both the last and extra test completion settings. Last test completion is nearly perfectly correlated with extra test completion performance. We hypothesize this is because the tasks are very similar, with the only difference being one additional test added to the context.

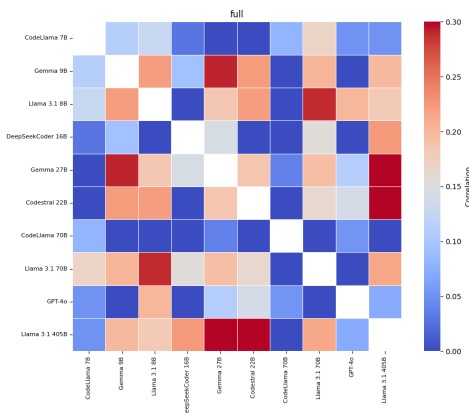

Figure 27: TESTGENEVALLITE full model correlation

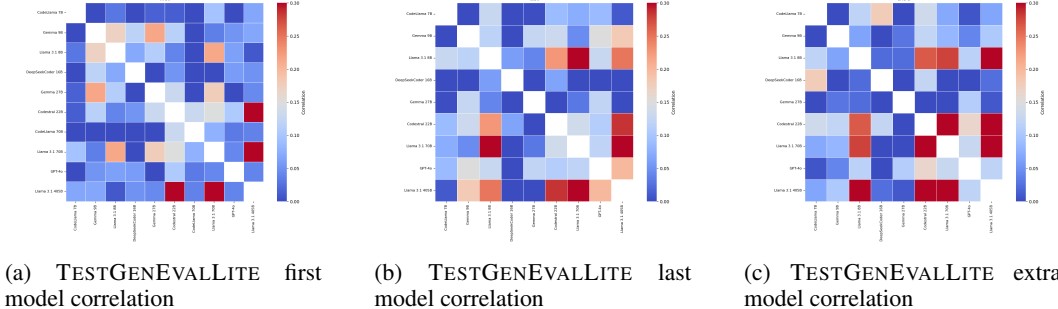

(a) TESTGENEVALLITE first model correlation

(b) TESTGENEVALLITE last model correlation

(c) TESTGENEVALLITE extra model correlation

Figure 28: Correlations for various test completion settings between all models in the TESTGENEVALLITE split. We omit CodeLlama 70B in settings where it has a pass@1 of 0.

Figure 27 shows the correlation the correlation of all models in their ability to solve the TESTGENEVAL full test generation setting. Similar to TESTGENEVAL, models in the same family (Llama 3.1, Gemma) exhibit the highest correlation between different sizes, larger models solve more of problems than smaller models, but still solve similar types of tasks.

Figure 28a, Figure 28b, and Figure 28c show the correlation of all models in their ability to solve the TESTGENEVALLITE first, last, and extra test completion settings. Similar to the full test generation, models in the same families are highly correlated in their ability to solve similar problems. Codestral is also more correlated with Llama 3.1 8B than other models for the extra test completion task.

### F.3 EFFECT OF SAMPLING ON COVERAGE AND PASS@K

Figure 29a and Figure 29b show how performance in the full test generation and first, last and extra test completion settings changes as more tests are sampled. For full test suite generation, we find that coverage at k gradually increases, with no noticeable plateau in the first 20 generations. The curve is slightly steeper at the beginning, but still looks linear as k approaches 20. Sampling 20 generations leads to approximately a 10% increase in overall coverage (coverage@20 is 23.3%).

For test completion, we find that pass@k is most substantial in the first 5 tests, but slowly increases after. Pass@100 performance is substantially higher than pass@1 and pass@5 performance, indicating models are capable of solving test completion tasks, but with many attempts. The plot curve

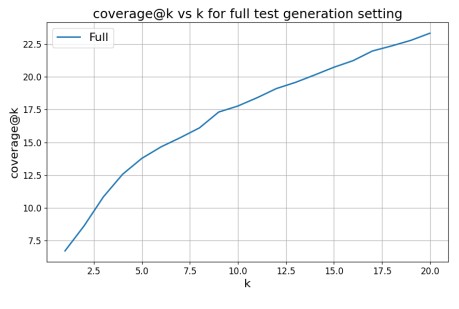 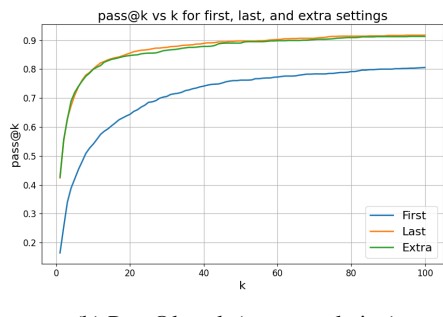

(a) Coverage@k vs k (full test generation)  (b) Pass@k vs k (test completion)

Figure 29: Effect of sampling more tests for all settings on Llama 3.1 8B. Coverage@k seems to gradually increase. Pass@k also increases more for lower k values and seems to plateau after k=20.

is steep initially, and then decays in value as more samples are added (first test completion plateaus around 80% pass@k and last and extra test completion plateaus around 90% pass@k).

### F.4  EFFECT OF CONTEXT ON COVERAGE AND PASS@5

We measure the effect of context length on both coverage for our full test generation setting and for pass@5 for our test completion setting.

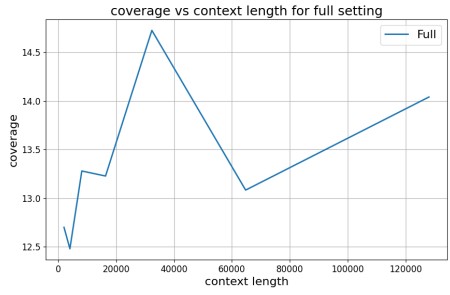 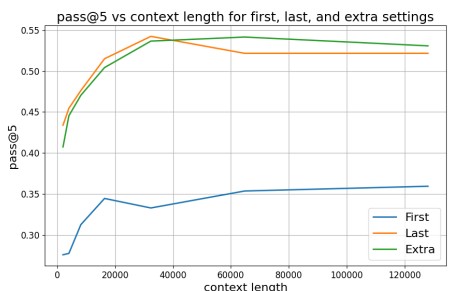

(a) Coverage by context length (full test generation)  (b) Pass@5 by context length (test completion)

Figure 30: Effect of context window for both our full setting (coverage of generated test suite) and for our test completion setting (pass@5 for our first, last and extra test completion settings). We find that context length generally helps test completion, however for test generation, even receiving parts of the file under test (measured by a lower context window) seems to be effective.

Figure 30a and Figure 30b show the effect of context length on both our coverage (full test generation) and on pass@5 (test completion). For full test suite generation, context does not help much; seeing part of a large file with many classes is enough to generate covering tests. Since tests can mock inputs in the partial test file, this partial context is sufficient to generate large test suite. However, for test completion context size helps significantly more. This is because to complete the next test in a file one needs knowledge of both the code under test and the existing test file. Full context ensures that the mapping between existing tests and the code under test is clear, improving performance of completing additional tests.

### F.5  QUANTITATIVE ERROR ANALYSIS

#### F.5.1  TESTGENEVAL

We bucket errors based off the logs from executing tests. Specifically for setting and generation we scrape the logs and extract the Python error class. Figure 31 shows the most frequent errors made by the best performing model for test completion, Codestral 22B: specifically assertion, no assertion,

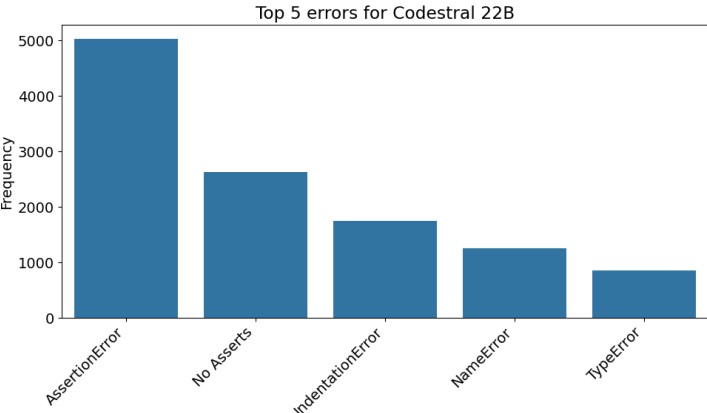

Figure 31: Different errors made on TESTGENEVAL by Codestral 22B (the best model on test completion). Despite high performance, Codestral struggles to reason about execution (large number of assertion and timeout errors).

timeout, indentation and name errors. As evidenced by the large number of assertion and timeout errors Codestral 22B struggles to reason about code execution. There are also cases where Codestral 22B generates tests that do not contain an assert and simply invoke the method under test. We filter these out and label such cases as an error.

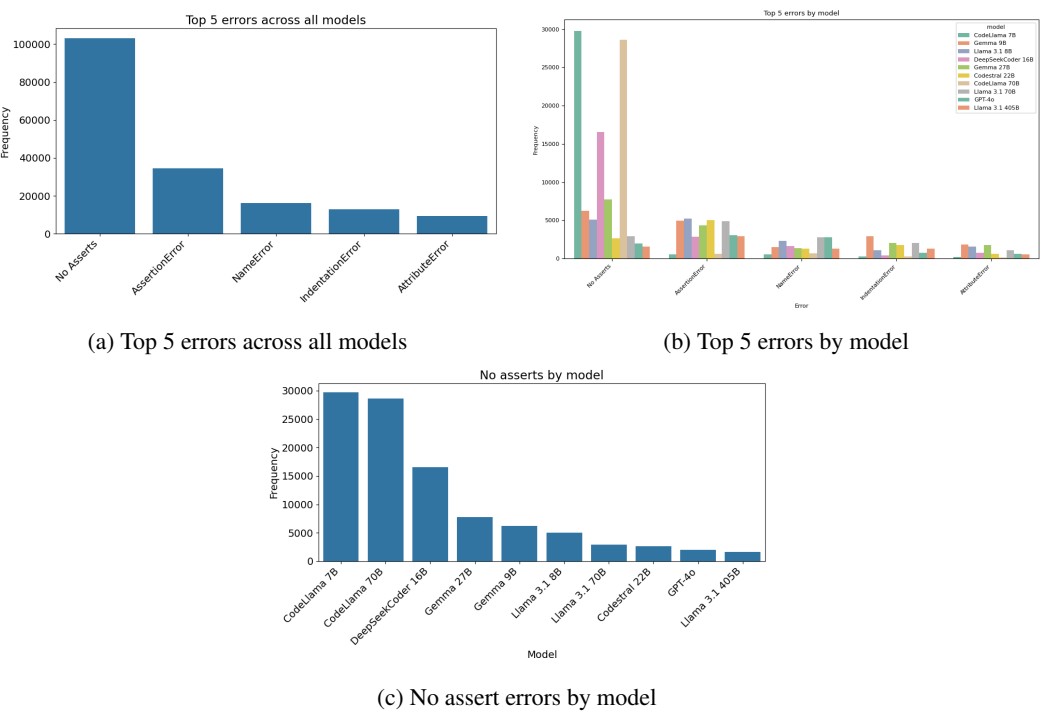

(a) Top 5 errors across all models          (b) Top 5 errors by model

(c) No assert errors by model

Figure 32: Common errors across all models. We find that the most common errors include not generating tests with asserts, followed by execution and hallucination related errors.

Figure 32a, Figure 32b, and Figure 32c show examples of the most common errors made by all models in TESTGENEVAL. We find that all models struggle the most with following the prompt, generating tests that lack assertions. After this error, the most common error types have to do with code execution (assertion, timeout errors), model hallucination (name error) and formatting (indentation error).

### F.5.2 TESTGENEVALLITE

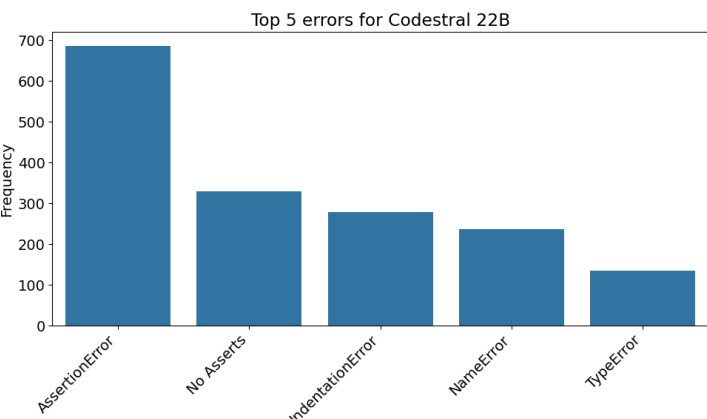

Figure 33: Different errors made on TESTGENEVALLITE by Codestral 22B. Results are similar to TESTGENEVAL.

Figure 33 shows the most frequent errors made by our best performing model Codestral 22B. Error counts remain the same as TESTGENEVAL.

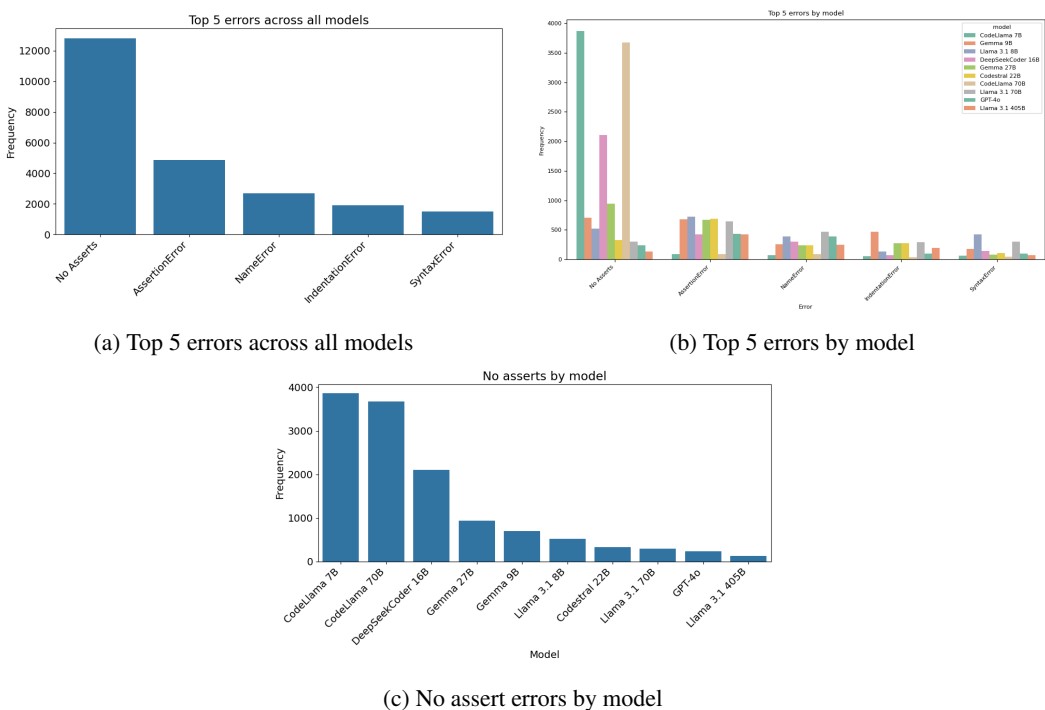

(a) Top 5 errors across all models

(b) Top 5 errors by model

(c) No assert errors by model

Figure 34: Common errors across all models. Results are similar to TESTGENEVAL.

Figure 34a, Figure 34b, and Figure 34c show examples of the most common errors made by all models in TESTGENEVALLITE. Similar to TESTGENEVAL, we find that all models struggle the most with following the prompt, generating tests that lack assertions and making both execution and hallucination errors.

### F.6 QUALITATIVE MODEL COMPARISON

We outline three cases where TESTGENEVAL discriminates between GPT-4o, CodeStral and Llama 3.1 405B (only one model suceeds in each of these examples).

#### F.6.1 EXAMPLE 1 - TEST SETUP (ONLY SOLVED BY GPT-4O)

```python
class QuerySet(AltersData):
    def __init__(self, model=None, query=None, ...):
        ...
    def repr(self):
        return "<%s %r>" % (self.__class__.__name__, data)
    ...
```

Listing 4: Query set class for managing data from database.

Our first example involves a QuerySet class that manages records returned from a database. The class has no database dependencies. The initialization takes a model and query, which can also not be set if no database is being used. This is hard to test because it involves either mocking or creating a complex model object.

```python
def test_queryset_repr(self):
    queryset = QuerySet(model=None)
    assertEqual(repr(queryset), "<QuerySet []>")
```

Listing 5: Cut GPT-4o generated test

GPT-4o instantiates the QuerySet with no parameters (the simplest possible version of the test). It then tests all code methods, with all of the generated tests passing on the code under test.

```python
from django.db import connection, models

def test_query_set_init(self):
    model = models.Model()
    qs = QuerySet(model=model)
    self.assertEqual(qs.model, model)
```

Listing 6: Cut Llama 3.1 405B generated test

This is not the case with both Llama 3.1 405B and Codestral 22B. Both models ignore the empty case entirely and instead hallucinate invocations or imports of models in Django (instead these models should have mocked the model object and tested the null case). All models failing to properly mock the class under test's dependencies lead to low coverage of the source file, despite GPT-4o generating passing tests all when other models fail.

#### F.6.2 EXAMPLE 2 - INCORRECTLY MOCKING OBJECTS (ONLY SOLVED BY LLAMA 3.1 405B)

```python
class MigrationAutodetector:
    def __init__(self, from_state, to_state, questioner=None):
        ...
        self.existing_apps = {app for app, model in
from_state.models}
    def changes(self, graph, trim_to_apps=None,
convert_apps=None, migration_name=None):
        ...
```

Listing 7: Migration detection class to automatically detect changes needed between project states.

Our second example involves a MigrationAutodetector class that takes in two project states and measures the changes between the states passed in to automatically detect code that should be migrated.

```
def test_changes():
    class Model(models.Model):
        pass
    ...
    changes = migration.changes(graph={}, trim_to_apps=None)
    self.assertEqual(len(changes), class="syntax-number">1)
```

Listing 8: Cut Llama 3.1 405B generated test

Llama 3.1 405B generates passing test for the changes method, providing an empty graph and model, meaning the only difference between the from and to states is the addition of the model. The assert passes, as the method only outputs one change.

```
def test_init():
    from_state = MagicMock(spec=ProjectState)
    to_state = MagicMock(spec=ProjectState)
    autodetector = MigrationAutodetector(self.from_state,
self.to_state, self.questioner)
    self.assertEqual(autodetector.from_state, self.from_state)
```

Listing 9: Cut GPT-4o generated test

GPT-4o attempts to mock the ProjectState class (passing instance checks of the ProjectState class). However, it misses the line in initialization where the models attribute of `from_state` is accessed. This leads the setup to fail, and the MigrationAutodetector initialization fails. Codestral 22B makes a similar error, where it incorrectly mocks the ProjectState class and misses the model attribute.

### F.6.3 EXAMPLE 3 - HANDLING CLASS DEPENDENCIES (ONLY SOLVED BY CODESTRAL 22B)

```
class FieldGetDbPrepValueMixin:
    ...
class Exact(FieldGetDbPrepValueMixin, BuiltinLookup):
    def process_rhs(self, compiler, connection):
        ...
```

Listing 10: Database classes to handle lookups and comparisons in queries.

Our final example deals with lookups and comparisons that could be applied when searching a database (for example exact match, greater than, less than, etc.). The file under test has many classes and subclasses such as `Exact`, which depends on `FieldGetDbPrepValueMixin`.

```
def test_exact_lookup():
    lhs = F('field')
    lookup = Exact(lhs, 'value')
    self.assertIn('%s', ...)
```

Listing 11: Cut GPT-4o generated test

GPT-4o and Llama 3.1 405B both hallucinate class invocations, rather than mocking or using the classes provided in context. F is not a correct class to pass as lhs (missing the required output field).

```
def test_exact():
    field = Field()
    lookup = Exact(field, 'value')
    self.assertEqual(lookup.lookup_name, 'exact')
```

Listing 12: Cut Codestral 22B generated test

Unlike GPT-4o and Llama 3.1 405B, Codestral 22B is able to understand the file under test. Codestral 22B both imports and instantiates valid field under test that is also imported in the file under test and correctly instantiates the Exact class that has a dependency on the field.

### F.7 QUALITATIVE ERROR ANALYSIS

We manually examine errors made by Codestral (the top model for our test completion task) and highlight examples for the most popular errors for TESTGENEVAL. Generally, models struggle with reasoning about execution (high amount of assertion errors) and hallucination (leads to timeout and value errors).

**Assertion Errors:** A common type of error is the model generating tests with incorrect asserts. This is challenging, as models often miss nuances in code execution and methods under test themselves.

```python
def prepare_import(path):
    """Given a filename this will try to calculate the python
path, checks the python path exists otherwise returns ''"""
...

# First test generation
def test_prepare_import():
    path = test_path / "simple_app.py"
    module_name = prepare_import(str(path))
    assert module_name == "simple_app"
```

Listing 13: Codestral first test method completion and code method

This example shows a case where the code under test requires setup (creating directories), but the model fails to add the required setup (either mocking the os methods or creating the required file in test path so that the import is recognized).

```python
def test_min_maxima_ratio():
    ...
    clust1 = OPTICS(min_samples=9,
min_maxima_ratio=0.001).fit(X)
    clust2 = OPTICS(min_samples=9,
min_maxima_ratio=0.01).fit(X)

    assert not np.array_equal(clust1.labels_, clust2.labels_)
```

Listing 14: Codestral last test method completion

An example where Codestral asserts that two objects that have identical labels are not equal. This is a case where Codestral misunderstands the code under test, which does not include min_maxima_ratio in the labels.

**No Assert Errors:** Another common type of error is no assert errors. We filter out generated tests that do not contain asserts.

```python
def test_all_world2pix_with_adaptive_and_detect_divergence():
    # Open test FITS file:
    fname = get_pkg_data_filename('data/j94f05bgq_flt.fits')
    ext = ('SCI', 1)
    h = fits.open(fname)
    w = wcs.WCS(h[ext].header, h)
...
```

Listing 15: Codestral extra test completion (no asserts)

```python
def test_tight_layout_h_pad_w_pad():
    fig, ax = plt.subplots(2,
2)
    fig.tight_layout(pad=1.0,
h_pad=2.0,
w_pad=3.0)
```

```
        plt.close(fig)
```
Listing 16: Codestral first test completion (no asserts)

**Timeout Errors:** Another common type of error is generating tests that lead to a degenerative state (infinite loops, timeout errors).

```
def test_server_handler_cleanup_headers(self):
    from django.core.servers.basehttp import ServerHandler

    class MockRequestHandler:
        class MockServer:
            pass

            self.server = self.MockServer()
            self.headers = {}

            pass
    ...
    server_handler.cleanup_headers()
    self.assertNotIn('Connection', server_handler.headers)
```
Listing 17: Codestral extra test method completion

This example shows a case where Codestral incorrectly mocks a request handler, by putting self.server = self.MockServer() in the class MockServer rather than in a __init__ method in MockRequestHandler. The cleanup_headers method requires that the headers are well formed to properly close the connection, and thus never actually closes the connection.

```
def test_get_traceback_data(self):
    request = RequestFactory().get('/')
    exc_type = TemplateDoesNotExist
    exc_value = TemplateDoesNotExist("Error message")
    tb = mock.Mock()
    reporter = ExceptionReporter(request, exc_type, exc_value,
tb)
    data = reporter.get_traceback_data()
    self.assertTrue(data['template_does_not_exist']
```
Listing 18: Codestral first test method completion

This shows a similar example with mocking. Because the mock will continuously return a non None object when called by get_traceback_data in a loop (tb = tb.tb_next), the model needs to mock the traceback correctly and ensure that non null is returned.

**Value Errors:** Another common type of error is generating tests that invoke functions with incorrect arguments (a value error) or missing some precondition to the function.

```
def test_timezone_aware(self):
    now = timezone.now()
    future = now + self.onehour
    future = timezone.localtime(future)
    self.assertEqual(timesince(now, future),
"class="syntax-number">1 hour")
    self.assertEqual(timeuntil(future, now),
"class="syntax-number">1 hour")
```
Listing 19: Codestral last test method completion

This example shows a case where Codestral misses an implementation detail of localtime in the code under test. Specifically, localtime cannot be applied on a naive datetime, instead a correct test would use the .localize method to convert a datetime object to an object with a timezone.

```
def test_get_paths(self):
    view = JavaScriptCatalog()
    paths = view.get_paths(['django.conf'])
    self.assertEqual(paths, [path.join(settings.BASE_DIR,
'django/conf/locale')])
    with self.assertRaises(ValueError):
        view.get_paths(['nonexistent_package'])
```

Listing 20: Codestral first test method completion

This shows a different example. Specifically, the input 'django.conf' is not a valid package that can be passed to the get_paths method. The code under test shows a way to get valid paths (invoking the apps.get_app_configs method, but instead of following this, the model hallucinates a non-existant package).

```
def test_get_paths(self):
    view = JavaScriptCatalog()
    paths = view.get_paths(['django.conf'])
    self.assertEqual(paths, [path.join(settings.BASE_DIR,
'django/conf/locale')])
    with self.assertRaises(ValueError):
        view.get_paths(['nonexistent_package'])
```

Listing 21: Codestral first test method completion

This shows a different example. Specifically, the input 'django.conf' is not a valid package that can be passed to the get_paths method. The code under test shows a way to get valid paths (invoking the apps.get_app_configs method, but instead of following this, the model hallucinates a non-existant package).

## F.8 FRONTIERS OF TESTGENEVAL

We also show some code snippets on the frontier of TESTGENEVAL. Specifically we show two examples of code snippets that no model can generate tests for and code snippets where only one model can generate tests.

**No Solve Cases:** Below are two cases where no model is capable of generating tests.

```
class Plane(GeometryEntity):
    def __new__(cls, p1, a=None, b=None, **kwargs):
        p1 = Point3D(p1, dim=class="syntax-number">3)
        if a and b:
            p2 = Point(a, dim=class="syntax-number">3)
            p3 = Point(b, dim=class="syntax-number">3)
            if Point3D.are_collinear(p1, p2, p3):
                raise ValueError('Enter three non-collinear
points')
            a = p1.direction_ratio(p2)
            b = p1.direction_ratio(p3)
            normal_vector = tuple(Matrix(a).cross(Matrix(b)))
        else:
            a = kwargs.pop('normal_vector', a)
            if is_sequence(a) and len(a) ==
class="syntax-number">3:
                normal_vector = Point3D(a).args
            else:
                raise ValueError(filldedent('''
                    Either provide class="syntax-number">3 3D
points or a point with a
```

```
                    normal vector expressed as a sequence of
length class="syntax-number">3'''))
            if all(coord.is_zero for coord in normal_vector):
                raise ValueError('Normal vector cannot be zero
vector')
        return GeometryEntity.__new__(cls, p1, normal_vector,
**kwargs)
    ...
```

Listing 22: No solve problem (class with many nested methods)

The above class is an example of a code file that no model is capable of generating tests for. This is because generating tests for such a class requires understanding complex execution (we only show one of many methods for brevity, but all methods involve complex execution sequences of methods). Furthermore, generating a unit test for such a class requires intricate knowledge of the code under test or complex mocking of all intermediate classes.

```
def write_latex(
    cosmology, file, *, overwrite=False, cls=QTable,
latex_names=True, **kwargs
):
    ...
    format = kwargs.pop("format", "latex")
    if format not in ("latex", "ascii.latex"):
        raise ValueError(f"format must be 'latex' or
'ascii.latex', not {format}")

    # Set cosmology_in_meta as false for now since there is no
metadata being kept
    table = to_table(cosmology, cls=cls,
cosmology_in_meta=False)

    cosmo_cls = type(cosmology)
    for name, col in table.columns.copy().items():
        param = getattr(cosmo_cls, name, None)
        if not isinstance(param, Parameter) or param.unit in
(None, u.one):
            continue
        # Get column to correct unit
        table[name] <<= param.unit

    # Convert parameter names to LaTeX format
    if latex_names:
        new_names = [_FORMAT_TABLE.get(k, k) for k in
cosmology.__parameters__]
        table.rename_columns(cosmology.__parameters__,
new_names)

    table.write(file, overwrite=overwrite, format="latex",
**kwargs)
```

Listing 23: No solve problem (dealing with IO/Files)

The above code file is another case where models fail to generate valid tests. The most common issue across models for this class is incorrectly testing the output file, as the method under test writes its output to a file. Checking this output requires intricate knowledge of the content written to the file, or complex mocking of IO operations. Ideally a model generating a unit test for this class would either need increased context, or proper mocking of file writing.

**One Solve Cases:** Below are two cases where only one model is capable of generating tests (frontier problems for TESTGENEVAL).

```
class RawModelIterable(BaseIterable):
    #Iterable that yields a model instance for each row from a
raw queryset.

class NamedValuesListIterable(ValuesListIterable):
    # Iterable returned by QuerySet.values_list(named=True)

class FlatValuesListIterable(BaseIterable):
    # Iterable returned by QuerySet.values_list(flat=True)
...
class QuerySet(AltersData):

# GPT-4o test generation

class QuerySetTests(TestCase):
    ...
    def test_queryset_getstate(self):
        state = self.queryset.__getstate__()
        self.assertIn(DJANGO_VERSION_PICKLE_KEY, state)
        self.assertEqual(state[DJANGO_VERSION_PICKLE_KEY],
django.__version__)

    def test_queryset_iter(self):
        self.assertEqual(list(iter(self.queryset)), [])
```

Listing 24: One solve problem (only GPT-4o)

This example highlights a case where in order to write tests for the main class (QuerySet) the model must understand a large number of classes defined in the code under test. GPT-4 successfully generates tests for methods that return these additional classes.

```
class RawModelIterable(BaseIterable):
    #Iterable that yields a model instance for each row from a
raw queryset.

class NamedValuesListIterable(ValuesListIterable):
    # Iterable returned by QuerySet.values_list(named=True)

class FlatValuesListIterable(BaseIterable):
    # Iterable returned by QuerySet.values_list(flat=True)
...
class QuerySet(AltersData):

# GPT-4o test generation

class QuerySetTests(TestCase):
    ...
    def test_queryset_getstate(self):
        state = self.queryset.__getstate__()
        self.assertIn(DJANGO_VERSION_PICKLE_KEY, state)
        self.assertEqual(state[DJANGO_VERSION_PICKLE_KEY],
django.__version__)

    def test_queryset_iter(self):
        self.assertEqual(list(iter(self.queryset)), [])
```

Listing 25: One solve problem (only GPT-4o)

This example highlights a case where in order to write tests for the main class (QuerySet) the model must understand a large number of classes defined in the code under test. GPT-4 successfully generates tests for methods that return these additional classes.

```python
class ModelOperation(Operation):
    def __init__(self, name):
        self.name = name

    @cached_property
    def name_lower(self):
        return self.name.lower()

    def references_model(self, name, app_label):
        return name.lower() == self.name_lower

    def reduce(self, operation, app_label):
        return super().reduce(operation, app_label) or
self.can_reduce_through(
            operation, app_label
        )

    def can_reduce_through(self, operation, app_label):
        return not operation.references_model(self.name,
app_label)

class CreateModel(ModelOperation):
    """Create a model's table."""
    ...

class DeleteModel(ModelOperation):
    """Drop a model's table."""
    ...

# Codestral 22B generated test

class TestModelOperations(unittest.TestCase):
    def setUp(self):
        self.app_label = 'test_app'
        self.model_name = 'TestModel'
        self.fields = [('id',
models.AutoField(primary_key=True))]
        self.options = {'managed': True}
        self.bases = (models.Model,)
        self.managers = []

    def test_create_model(self):
        operation = CreateModel(self.model_name, self.fields,
self.options, self.bases, self.managers)
        self.assertEqual(operation.name, self.model_name)
        self.assertEqual(operation.fields, self.fields)
        self.assertEqual(operation.options, self.options)
        self.assertEqual(operation.bases, self.bases)
        self.assertEqual(operation.managers, self.managers)
    ...
```

Listing 26: One solve problem (only Codestral 22B)

This shows another example of a one solve problem. The challenge here is the subclassing of the CreateModel class. Codestral is able to successfully invoke these subclasses and come up with valid asserts.

# G  STATISTICAL TESTS

We run pairwise comparisons for both TESTGENEVAL and TESTGENEVALLITE splits of our data.

## G.1  TESTGENEVAL

Table 12: 95% confidence interval for pass@1 along with noise related statistics for each of TEST-GENEVAL settings.

| Setting | 95% Int. | No Solve | Tau- | Sig-Noise |
|---|---|---|---|---|
| Test generation | 3.1% | 4.7% | 6.4% | 2.70 |
| Extra test completion | 3.5% | 12.0% | 11.2% | 1.21 |
| First test completion | 3.7% | 25.9% | 7.8% | 1.87 |
| Last test completion | 4.4% | 10.5% | 10.4% | 1.27 |

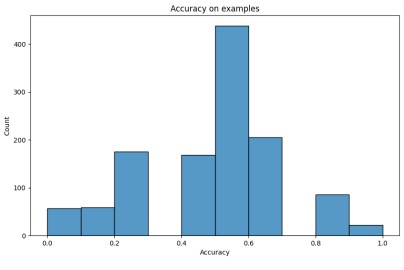

(a) TESTGENEVAL acc by problem

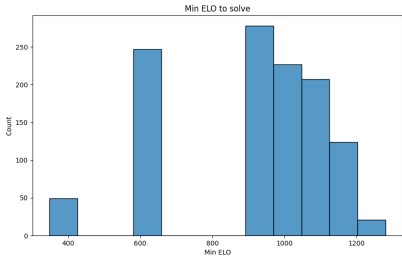

(b) Min elo to solve problems

Figure 35: Problems by average model accuracy and min elo required to solve problems for full test generation setting

Figure 35a and Figure 35b show the average model accuracies by problem, along with the min elo required to solve each problem for our full test generation setting.

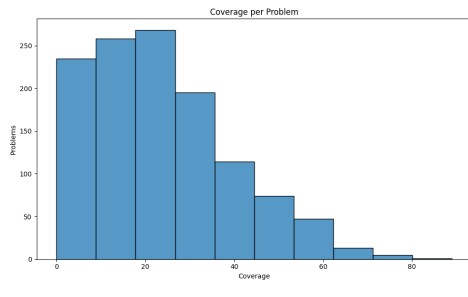

(a) TESTGENEVAL cov by problem

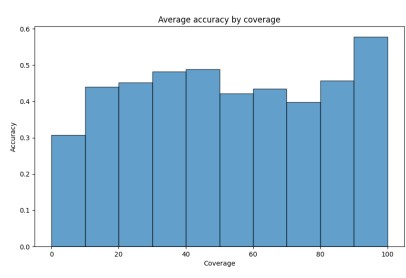

(b) TESTGENEVAL acc per cov

Figure 36: Problems by average model coverage and average accuracy per coverage

Figure 37a and Figure 37b show the average model accuracies by problem, along with the min elo required to solve each problem for our extra test completion setting.

Figures Figure 38a and Figure 38b show the average model accuracies by problem, along with the min elo required to solve each problem for our extra test completion setting.

Figure 39a and Figure 39b show the average model accuracies by problem, along with the min elo required to solve each problem for our first test completion setting.

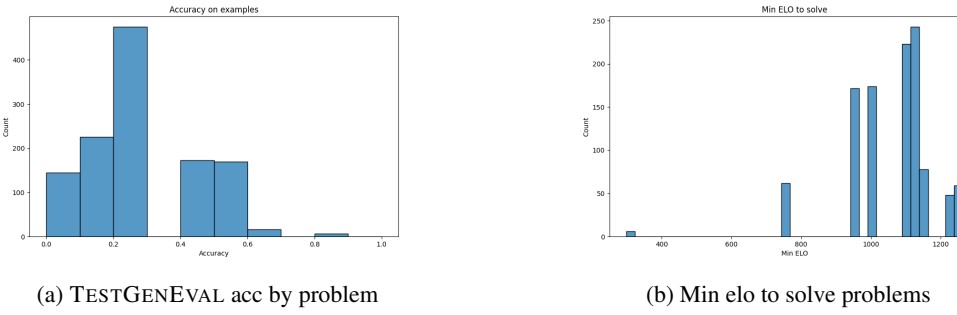

(a) TESTGENEVAL acc by problem

(b) Min elo to solve problems

Figure 37: Problems by average model accuracy and min elo required to solve problems for extra test completion generation setting

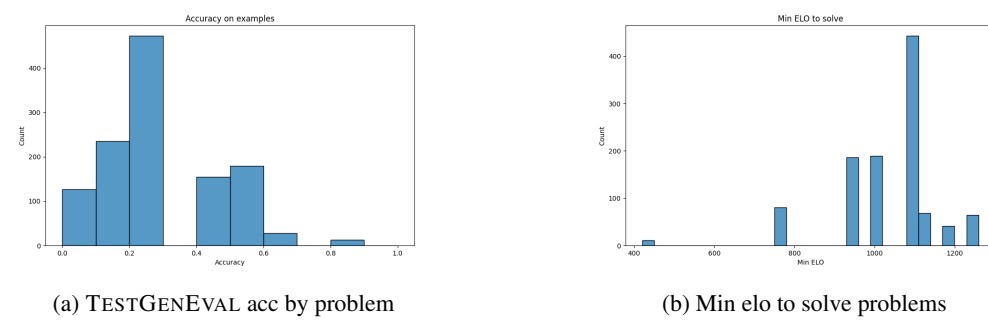

(a) TESTGENEVAL acc by problem

(b) Min elo to solve problems

Figure 38: Problems by average model accuracy and min elo required to solve problems for last test completion setting

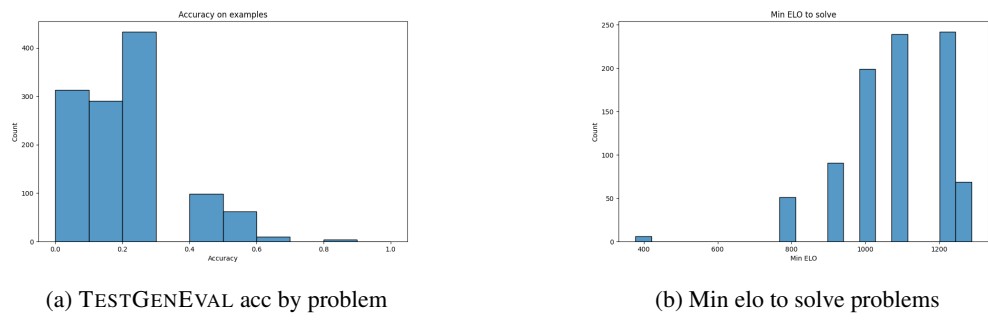

(a) TESTGENEVAL acc by problem

(b) Min elo to solve problems

Figure 39: Problems by average model accuracy and min elo required to solve problems for first test completion setting

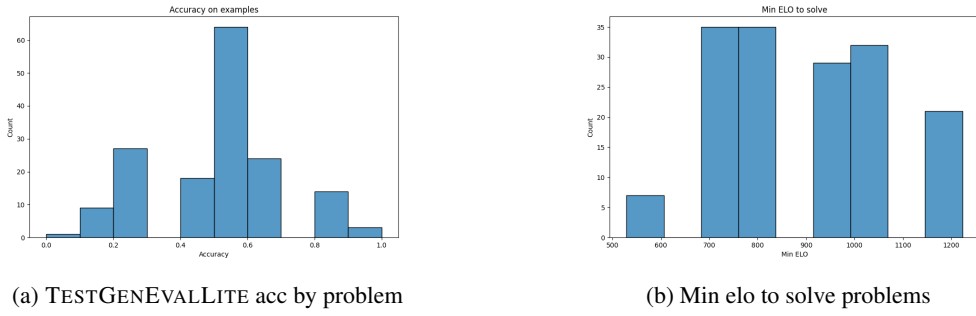

(a) TESTGENEVALLITE acc by problem

(b) Min elo to solve problems

Figure 40: Problems by average model accuracy and min elo required to solve problems for full test generation setting

Table 13: Model performance for the full test suite generation setting. Coverage refers to the coverage of generated tests.

| Model | Coverage | Win Rate | Elo |
|---|---|---|---|
| GPT-4o | 35.2% | 79.6% | 1280.6 |
| Llama 3.1 405B | 35.0% | 72.8% | 1221.2 |
| Codestral 22B | 33.0% | 65.8% | 1161.2 |
| Llama 3.1 70B | 30.6% | 66.3% | 1168.8 |
| Gemma 27B | 30.1% | 69.8% | 1196.5 |
| DeepSeekCoder 16B | 28.2% | 53.4% | 1086.1 |
| Gemma 9B | 20.2% | 41.2% | 996.7 |
| Llama 3.1 8B | 14.1% | 27.1% | 899.1 |
| CodeLlama 70B | 7.0% | 7.3% | 641.9 |
| CodeLlama 7B | 1.3% | 1.4% | 347.8 |

Table 14: Model performance for the first test completion setting. Pass@1 refers to if any of the generated tests pass.

| Model | Pass@1 | Win Rate | Elo |
|---|---|---|---|
| Codestral 22B | 38.3% | 80.4% | 1288.1 |
| Llama 3.1 405B | 32.1% | 73.7% | 1231.1 |
| GPT-4o | 31.9% | 71.8% | 1220.4 |
| Llama 3.1 70B | 19.3% | 52.6% | 1085.9 |
| DeepSeekCoder 16B | 18.6% | 51.2% | 1082.6 |
| Llama 3.1 8B | 14.4% | 41.7% | 1025.1 |
| Gemma 27B | 12.7% | 37.6% | 1003.0 |
| Gemma 9B | 8.4% | 25.9% | 912.0 |
| CodeLlama 7B | 4.2% | 13.9% | 774.6 |
| CodeLlama 70B | 0.5% | 1.6% | 377.2 |

## G.2 TESTGENEVALLITE

Figure 40a and Figure 40b show the average model accuracies by problem, along with the min elo required to solve each problem for our full test generation setting.

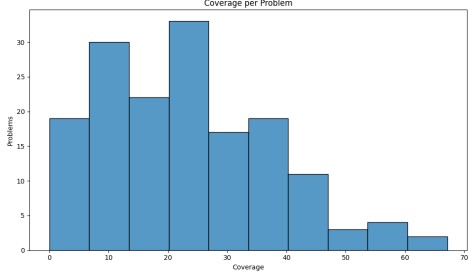

(a) TESTGENEVALLITE cov by problem

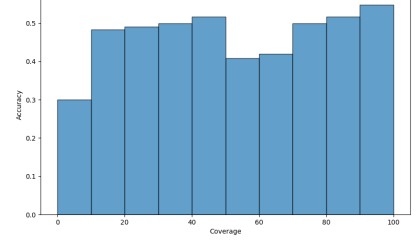

(b) TESTGENEVALLITE acc per cov

Figure 41: Problems by average model coverage and average accuracy per coverage

Figure 42a and Figure 42b show the average model accuracies by problem, along with the min elo required to solve each problem for our extra test completion setting.

Figure 43a and Figure 43b show the average model accuracies by problem, along with the min elo required to solve each problem for our extra test completion setting.

Figure 44a and Figure 44b show the average model accuracies by problem, along with the min elo required to solve each problem for our first test completion setting.

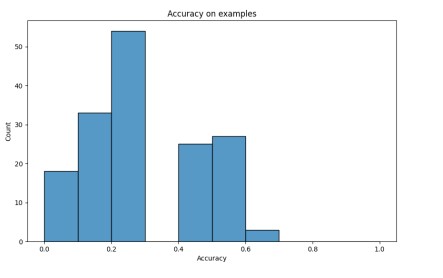

(a) TESTGENEVALLITE acc by problem

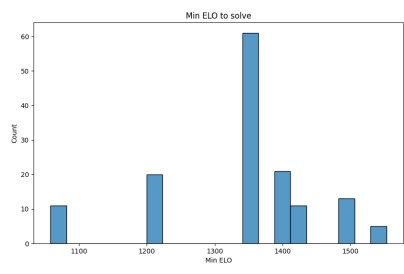

(b) Min elo to solve problems

Figure 42: Problems by average model accuracy and min elo required to solve problems for extra test completion generation setting

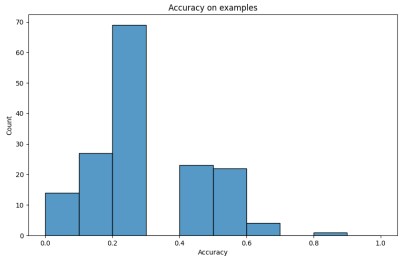

(a) TESTGENEVALLITE acc by problem

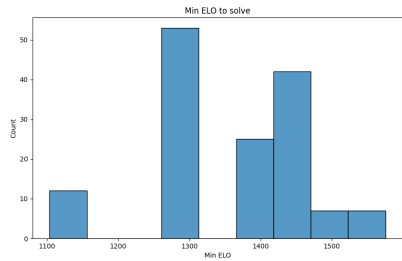

(b) Min elo to solve problems

Figure 43: Problems by average model accuracy and min elo required to solve problems for last test completion setting

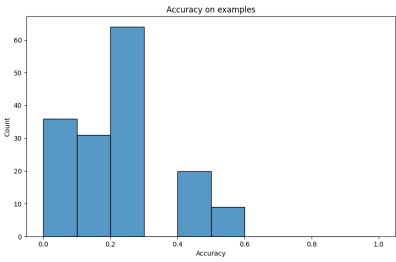

(a) TESTGENEVALLITE acc by problem

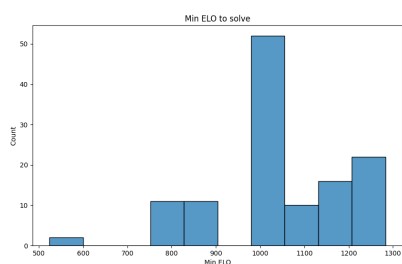

(b) Min elo to solve problems

Figure 44: Problems by average model accuracy and min elo required to solve problems for full test generation setting

Table 15: Model performance for the last test completion setting. Pass@1 refers to if any of the generated tests pass.

| Model | Pass@1 | Win Rate | Elo |
| --- | --- | --- | --- |
| Codestral 22B | 50.4% | 79.1% | 1258.6 |
| Llama 3.1 405B | 42.6% | 72.1% | 1195.1 |
| Llama-3.1 70B | 35.0% | 62.1% | 1122.9 |
| GPT-4o | 32.6% | 58.0% | 1103.5 |
| Gemma 27B | 32.2% | 57.3% | 1105.4 |
| Llama 3.1 8B | 32.0% | 57.6% | 1096.5 |
| Gemma 9B | 21.4% | 40.8% | 994.0 |
| DeepSeekCoder 16B | 17.0% | 33.4% | 944.7 |
| CodeLlama 7B | 6.9% | 13.9% | 758.6 |
| CodeLlama 70B | 0.9% | 2.2% | 420.7 |

Table 16: Model performance for the extra test completion setting. Pass@1 refers to if any of the generated tests pass.

| Model | Pass@1 | Win Rate | Elo |
| --- | --- | --- | --- |
| Codestral 22B | 48.3% | 77.9% | 1262.1 |
| Llama 3.1 405B | 42.4% | 72.9% | 1216.4 |
| Llama 3.1 70B | 36.4% | 65.6% | 1160.7 |
| Llama 3.1 8B | 31.8% | 59.0% | 1121.8 |
| Gemma 27B | 31.7% | 57.8% | 1127.3 |
| GPT-4o | 30.4% | 56.5% | 1111.2 |
| Gemma 9B | 18.9% | 38.2% | 998.5 |
| DeepSeekCoder 16B | 15.5% | 32.2% | 958.2 |
| CodeLlama 7B | 5.2% | 11.6% | 744.5 |
| CodeLlama 70B | 0.5% | 1.0% | 299.2 |

Table 17: 95% confidence interval along with noise related statistics for each of TESTGENEVAL-LITE settings.

| Setting | 95% Int. | No Solve | Tau- | Noise |
| --- | --- | --- | --- | --- |
| Test generation | 5.6% | 0.6% | 6.2% | 1.16 |
| Extra test completion | 6.9% | 11.2% | 11.9% | 0.31 |
| First test completion | 5.6% | 22.5% | 6.9% | 0.87 |
| Last test completion | 7.5% | 8.8% | 10.0% | 0.13 |

Table 18: Model performance for the full test suite generation setting. Coverage refers to the coverage of generated tests.

| Model | Coverage | Win Rate | Elo |
| --- | --- | --- | --- |
| Llama 3.1 405B | 35.3% | 77.3% | 1245.2 |
| Codestral 22B | 34.9% | 69.1% | 1176.7 |
| GPT-4o | 32.9% | 74.6% | 1223.3 |
| Llama 3.1 70B | 31.3% | 66.8% | 1171.7 |
| Gemma 27B | 29.4% | 64.8% | 1148.5 |
| DeepSeekCoder 16B | 27.6% | 48.6% | 1045.2 |
| Gemma 9B | 20.2% | 38.1% | 968.0 |
| Llama 3.1 8B | 12.4% | 17.1% | 783.9 |
| CodeLlama 70B | 8.0% | 10.9% | 708.8 |
| CodeLlama 7B | 2.1% | 4.1% | 528.6 |

Table 19: Model performance for the first test completion setting. Pass@1 refers to if any of the generated tests pass.

| Model | Pass@1 | Win Rate | Elo |
|---|---|---|---|
| Codestral 22B | 43.1% | 81.7% | 1283.1 |
| GPT-4o | 36.2% | 72.7% | 1212.3 |
| Llama 3.1 405B | 33.8% | 71.7% | 1195.2 |
| Llama 3.1 70B | 21.9% | 53.6% | 1080.2 |
| DeepSeekCoder 16B | 19.4% | 48.9% | 1053.7 |
| Gemma 27B | 16.2% | 42.7% | 1012.9 |
| Llama 3.1 8B | 14.4% | 38.3% | 987.5 |
| CodeLlama 7B | 6.9% | 20.2% | 829.0 |
| Gemma 9B | 6.9% | 19.3% | 822.1 |
| CodeLlama 70B | 1.2% | 4.0% | 524.1 |

Table 20: Model performance for the last test completion setting. Pass@1 refers to if any of the generated tests pass. We omit CodeLlama 70B because it is not able to complete any tests on first generation.

| Model | Pass@1 | Win Rate | Elo |
|---|---|---|---|
| Codestral 22B | 51.2% | 79.2% | 1575.4 |
| Llama 3.1 405B | 40.0% | 68.0% | 1478.3 |
| Llama 3.1 70B | 37.5% | 64.1% | 1453.6 |
| Llama 3.1 8B | 36.9% | 63.9% | 1453.5 |
| Gemma 27B | 33.1% | 57.6% | 1424.4 |
| GPT-4o | 28.1% | 51.0% | 1373.3 |
| DeepSeekCoder 16B | 20.6% | 39.9% | 1310.2 |
| Gemma 9B | 19.4% | 36.5% | 1277.5 |
| CodeLlama 7B | 7.5% | 16.0% | 1103.5 |
| CodeLlama 70B | 0.0% | 0.0% | -2449.8 |

Table 21: Model performance for the extra test completion setting. Pass@1 refers to if any of the generated tests pass. We omit CodeLlama 70B because it is not able to complete any tests on first generation.

| Model | Pass@1 | Win Rate | Elo |
|---|---|---|---|
| Codestral 22B | 48.1% | 78.1% | 1552.9 |
| Llama 3.1 405B | 42.5% | 72.6% | 1501.1 |
| Llama 3.1 70B | 41.9% | 71.4% | 1494.1 |
| Llama 3.1 8B | 33.8% | 60.3% | 1419.8 |
| Gemma 27B | 31.9% | 56.4% | 1411.4 |
| GPT-4o | 26.2% | 48.8% | 1350.0 |
| Gemma 9B | 25.6% | 47.8% | 1349.1 |
| DeepSeekCoder 16B | 13.8% | 28.6% | 1221.7 |
| CodeLlama 7B | 6.9% | 13.5% | 1058.2 |
| CodeLlama 70B | 0.0% | 0.0% | -2358.4 |

# H LIMITATIONS

**Variation due to prompt and temperature** Model performance is highly dependant on the prompt and temperature used Gao et al. (2024); Wei et al. (2023). A potential limitation of our work is that we primarily focused on 0-shot performance, asking each model to generate the entire test suite or to complete a test given the code under test. To mitigate this we provide our prompt in Appendix C and also provide the model with the correct imports to the code under test to enable prompted models to generate the test suite successfully.

**Overfitting to SWEBench repositories:** Another potential limitation is that our benchmark is adapted from SWEBench and as a result risks models overfitting to this specific dataset. Currently, this does not seem to be a major issue, as model performance is low across the board. Even once models can achieve high coverage, a significantly harder task of achieving high mutation score (actually catching synthetic bugs introduced into the code under test). These multiple levels of difficulty and numerous tasks help mitigate the risk of a model overfitting to any one task specifically.

**Data contamination:** There is also a risk of data contamination in the pretraining data of models. To further understand data contamination, we measure perplexity of 10 randomly selected tests in TESTGENEVAL for Llama 3.1 8B and common frequent, and non recent code from GitHub with similar lengths. We find that the perplexity of this common GitHub code is lower than the 10 tests from TESTGENEVAL (1.6 vs 2.0), indicating that data is unlikely to be contaminated. This is further supported by the low performance of all models on TESTGENEVAL across the board.

**File level context:** TESTGENEVAL currently works at the file level. This means that the context given to the model is the source file, imports and for the test completion setting a partial test file. While in many cases, this context is sufficient sometimes there are cross file dependencies that are not captured, making the context provided to the model limited. However, we believe this is not a major limitation, as models can mock objects which they lack context for. Furthermore, expanding to repository level context is out of context window for most state-of-the-art models.

**Additional baselines:** One other limitation is the lack of agentic baselines. This might be a promising future work direction in overcoming some of the limitations of the context windows of current models, and could potentially perform better than baselines showed. However, the benchmark is still highly valuable to the community, as the first file level test generation benchmark.

**Compute cost of mutation score:** One other limitation is the compute cost of computing mutation score. Each synthetic bug we introduce to the code under test, requires an additional test suite execution. However, our results show that coverage and mutation score are highly correlated. Setting a timeout of one hour per mutation testing run, we only timeout on 20% of files, and get an average uncertainty of 1.1%. We provide an option to run TESTGENEVAL without mutation, enabling those who lack compute to still benefit from TESTGENEVAL.

**Misc Limitations:** Quantitative metrics such as code coverage and mutation score can approximate the quality of generated tests, however they do not perfectly measure the quality of generated tests. For example, a test could be very hard to read, while still achieving high coverage and mutation score. Additionally, all test generation benchmarks assume the code under test is correct. Generated tests may fail on the code under test, while exposing bugs in the code under test (a phenomenon known as the oracle problem). Despite this, current test generation approaches are still useful in catching regressions or bugs introduced in future versions of the code under test.

