# OpenReview forum: "TestGenEval: A Real World Unit Test Generation and Test Completion Benchmark"
_ICLR.cc/2025/Conference — ICLR 2025 Poster_

### Official Review · Reviewer_4d8B · 2024-10-26

**Soundness:** 3
**Presentation:** 2
**Contribution:** 3
**Rating:** 6
**Confidence:** 4

**Summary:**

This paper introduces TestGenEval,
a novel benchmark designed to evaluate neural test generation models on real-world Python projects.
TestGenEval is built upon 11 popular Python repositories selected from the patch generation benchmark SWE-Bench.
It assesses neural test generation from two key perspectives: test generation from scratch and test completion based on some existing test functions.
Both aspects, particularly test completion, are valuable for real-world software engineering applications, such as IDE plugins.
The authors evaluate several popular LLMs on TestGenEval,
pointing out that even the best-performing model GPT-4o struggles with achieving high code coverage.

**Strengths:**

* A standard, real-world project-based benchmark for test generation is crucial and useful for advancing software testing research.
* TestGenEval is the first benchmark to incorporate mutation score as a metric, a commonly used measure in software engineering to assess the robustness of test cases.
* It offers Docker images instrumented with coverage and mutation score, ensuring the reusability of the artifacts.
* TestGenEval is also the first benchmark to evaluate file-level test completion, which aligns well with the real-world software development workflows.
* The authors perform a comprehensive quantitative analysis of the results, demonstrating the correlations and effects of various components within the benchmark.

**Weaknesses:**

* This paper lacks details on mutation score calculation.
  The authors do not provide specifics on how mutation score is measured in the benchmark.
  In traditional software engineering, mutation testing is creating mutants of the code under test to evaluate the robustness of the test cases in detecting these modifications.
  However, the paper does not clarify the process used to construct program mutants for test generation,
  which is a non-trivial task that requires further clarification.


* The authors claim that real-world unit testing involves reasoning over complex *files*,
  generating tests for a given *class under test*.
  The authors also point out that the other related benchmark only measure the individual test method rather than an entire test suite is a weakness.
  This claim is not well supported with reference nor evidence,
  and is not commonly adopted by the software engineering community.
  Unit testing aims to validate the correctness of the smallest building block (and hence, an "unit") in a complex software project.
  In modern software engineering, the "units" of software are often functions in the codebase.
  If these functions are tested/verified to be correct, then the composition should be correct without further testing.
  Therefore, the best practices often suggest to write functions that are independent and de-coupled.
  The more recent and popular programming languages, for example Rust, often abandon class and favor function composition
  over inheritance.
  Are these languages not related to "real-world unit testing"?

* The authors claim that real-world unit testing requires reasoning over complex *files* and generating tests for a given *class under test*.
  They argue that other benchmarks' focus on individual test methods,
  rather than entire test suites, is a weakness.
  However, this claim is not well-supported by references or evidence and is not commonly adopted by the software engineering community.

  Unit testing aims to validate the smallest building blocks of software, or so-called "units,"
  which are often functions rather than entire classes.
  In modern software engineering, if individual functions are tested and verified to be correct,
  the composition of them is expected to be correct without additional testing.
  Consequently, best practices emphasize writing independent and de-coupled functions.

  Furthermore, more recent programming languages, such as Rust,
  favor function composition over inheritance, reflecting a shift away from class-based design.
  This raises the question of whether these languages fall outside the scope of "real-world unit testing" as claimed by the authors,
  despite their increasing relevance in software development.

* The lack of quantitative analysis results in the main text is a significant weakness.
  While the authors conduct five different quantitative analyses, all results are placed in the Appendix.
  If an analysis is mentioned in the main text, it would be more effective to present the corresponding results alongside the discussion.
  For space consideration, the authors could include only the most important analyses and results in the main text,
  while moving less critical details to the Appendix.
  This approach should also be applied to qualitative analyses for better readability and impact.


* Minor writing issue: the authors cite all references without using parentheses, regardless of context.
  In some cases, using `\citep` instead of `\cite` would enhance the reading experience by making the citations more contextually appropriate.
  Adjusting this would improve the flow and clarity of the text.

**Questions:**

Pleased address the concerns in **Weakness**.

---

> ### Author Response · Authors · 2024-11-17
>
> **…details on mutation score calculation…**
>
> Thank you for pointing this out. We add more details and references on how the mutants are generated in Appendix D.2.1.
>
> **…lack of quantitative analysis results in the main text is a significant weakness…**
>
> We agree with your feedback and made these updates in the revised paper version. We refactored the most important parts of our quantitative analysis to be part of the main paper and kept the remaining analysis in Appendix.
>
> **…the best practices often suggest to write functions that are independent and de-coupled…**
>
> This is true, however to effectively generate unit tests one needs knowledge of the entire file under test. Developers organize unit tests at the file level because it makes sense to have common setup (i.e. mocking objects for a file) followed by tests for individual methods in the file. Additionally, unit test frameworks such as JUnit generally expect generated tests to be at the file level; it is challenging to run individual tests in most frameworks, as people usually execute tests at the file level. Choosing to model our benchmark at the file level allows us to construct tasks that more closely resemble automated quality assurance, for example generating an entire test suite from code under test or adding to an existing complex test suite. LLM approaches that target test file generation outperform those at the method level (see Rao et. al. 2023 in comparison to Nie et. al. 2022), indicating that this file level context is essential for performance.
>
> **… Rust…whether these languages fall outside the scope of "real-world unit testing" as claimed by the authors…**
>
> We argue that Rust would still fall into our paradigm of generating tests at the file level. In Rust a file would be organized as a series of functions, thus if TestGenEval was extended to Rust, it would measure the ability of LLMs to test functions in a given Rust file. We changed the wording from *class* under test to *file* under test to improve clarity.
>
> **…using \citep instead of \cite…**
>
> Thank you for pointing this out. We agree with your comment and use \citep when appropriate.

---

> > ### Comment · Reviewer_4d8B · 2024-11-20
> >
> > Thanks for addressing my concerns and providing more details on mutation testing. I have raised my rating to 6.

---

### Official Review · Reviewer_FhSJ · 2024-10-31

**Soundness:** 3
**Presentation:** 3
**Contribution:** 4
**Rating:** 6
**Confidence:** 4

**Summary:**

The theme of this paper is the introduction and evaluation of a large-scale benchmark named TESTGENEVAL, which is designed to measure the performance of test generation and test completion.

TESTGENEVAL is constructed based on the SWEBench dataset and includes 68,647 test cases from 1,210 code and test file pairs across 11 well-maintained Python repositories.

This benchmark aims to fill the gap in existing code generation language model (LLM) benchmarks that focus less on software testing, despite the strong correlation between well-tested software and effective bug detection.

TESTGENEVAL covers tasks such as initial test authoring, test suite completion, and code coverage improvement, aiming to comprehensively assess test generation performance.

The paper also evaluates several popular models with parameter sizes ranging from 7B to 405B and provides a detailed analysis of TESTGENEVAL's contribution to the evaluation of test generation performance.

**Strengths:**

1. Continuous Improvement of Test Suites: TESTGENEVAL supports not only the generation of test suites from scratch but also the completion of existing test suites, which is particularly important for test optimization in continuous integration and continuous deployment (CI/CD) processes.

2. Evaluating and Comparing Different Models: TESTGENEVAL provides a standardized environment to evaluate and compare various code generation models, including both open-source and proprietary models, helping researchers and practitioners understand the performance of different models in real-world software testing tasks.

3. Test Generation for Real-World Use Cases: TESTGENEVAL is built based on real-world projects, meaning it is closer to actual development testing needs rather than being limited to academic research or toy problems.

**Weaknesses:**

1. Dependence on Prompts and Temperature Settings:
Model performance is highly dependent on the prompts and temperature parameters used. The paper focuses mainly on 0-shot performance, asking each model to generate an entire test suite or complete a test given the code under test, which may limit the model's performance.

2. Risk of Data Contamination:
There is a risk of data contamination in the pre-training data of models. Although the paper suggests that data contamination is unlikely by comparing the perplexity of tests in TESTGENEVAL with common GitHub code of similar length, it remains a potential issue.
In addition, since TESTGENEVAL is adapted from the SWEBench dataset, there is a risk of models overfitting to this specific dataset.

3. Computational Cost:
The computational cost of calculating mutation scores is high. Each synthetic bug introduced to the code under test requires an additional test suite execution.

4. Assumptions of Test Generation:
All test generation benchmarks assume that the code under test is correct. Generated tests may fail on the code under test while exposing bugs in the code under test (a phenomenon known as the oracle problem).

**Questions:**

1. I would like to know what is the cost involved in executing TestGenEval? Including length of prompt, required GPU memory, running time and more cost analysis?

2. How do the authors ensure the quality of the dataset, including but not limited to possible leakage of data and semantic correctness of the given code?

---

> ### Author Response · Authors · 2024-11-17
>
> **I would like to know what is the cost involved in executing TestGenEval? Including length of prompt, required GPU memory, running time and more cost analysis?**
>
> We generate ~11M tokens for TestGenEval and ~1.5M tokens for TestGenEvalLite (including both our test suite generation and test completion tasks). Runtime for TestGenEval is typically between 12-24 hours for the full benchmark (depending on model capability), across 16 CPUs and 2-4 hours for TestGenEvalLite across 16 CPUs. Running without mutation score significantly reduces runtime to 2-4 hours for TestGenEval and approximately 10-30 minutes for TestGenEvalLite for the same 16 CPU setup. For all settings, we use the full model context for TestGenEval (typically 128k tokens), however it is possible to run with a 32k context window with minimal performance differences (see Appendix F.4 for full details). Note that executing TestGenEval is significantly less costly than executing SWEBench (fixes for SWEBench often involve understanding multiple large files and running large trajectories, while TestGenEval focuses on generating test suites and test cases for a single file under test).
>
> **How do the authors ensure the quality of the dataset, including but not limited to possible leakage of data and semantic correctness of the given code?**
>
> To answer the data leakage question, we conducted a thorough experiment in line with existing papers on data leakage (https://arxiv.org/pdf/2404.18824, https://arxiv.org/pdf/2309.10677), measuring both n-gram accuracy and perplexity of SWEBenchLite (lite split of SWEBench and what TestGenEvalLite is based on) compared to popular software engineering benchmarks, and GitBug-Java (a benchmark of recent bugs that should not be leaked).
>
> Below are links to each dataset we compared against:
>
> Defects4J: https://github.com/rjust/defects4j
>
> BugsInPy: https://github.com/soarsmu/BugsInPy
>
> GitBug-Java: https://github.com/gitbugactions/gitbug-java
>
> SWEBenchLite: https://github.com/princeton-nlp/SWE-bench/tree/main/swebench
>
>
> Perplexity of different models:
> | **Model**         | **Defects4J** | **BugsInPy** | **GitBug-Java** | **SWEBenchLite** |
> |--------------------|---------------|--------------|-----------------|------------------|
> | Llama 3.1 8B       | 2.04          | 2.44         | 2.33            | 2.41             |
> | Llama 3.1 70B      | 1.76          | 1.98         | 2.07            | 1.93             |
> | CodeLlama 7B       | 1.58          | 1.81         | 1.68            | 1.80             |
>
> 5-gram match of different models:
>
> | **Model**         | **Defects4J** | **BugsInPy** | **GitBug-Java** | **SWEBenchLite** |
> |--------------------|---------------|--------------|-----------------|------------------|
> | Llama 3.1 8B       | 0.44          | 0.38         | 0.38            | 0.34             |
> | Llama 3.1 70B      | 0.51          | 0.47         | 0.45            | 0.47             |
> | CodeLlama 7B       | 0.64          | 0.54         | 0.58            | 0.53             |
>
> As we can see from both tables, the 5-gram match and perplexity of SWEBenchLite is relatively similar to GitBug-Java (a recent unleaked benchmark). For an older and likely leaked benchmark such as Defects4J, we can observe much greater differences in perplexity and 5-gram match, indicating that data contamination is unlikely to be an issue for TestGenEval at the moment.
>
> For semantic correctness of code, we measure both coverage and mutation score. While it is possible to achieve 100% code coverage by just invoking all lines in the method under test and asserting true, it is only possible to achieve high mutation score if the generated tests can discriminate between buggy and fixed code. The need to check more than syntactic correctness of generated solutions was a major motivation for including mutation score (a strong execution metric) as one of our evaluation metrics.

---

> ### Author Response · Authors · 2024-11-17
>
> **…Dependence on Prompts and Temperature Settings…**
>
> We agree that other settings than 0-shot performance would be relevant (and list this as a limitation in Appendix H). For our tasks, few-shot generation or agents could also be evaluated and we release all our code and containers to allow us to evaluate other methods on our benchmark easily. However, we also believe that comparing popular models for zero-shot generation is very relevant as it is close to the setting users often experience in chat applications and many programming tools.
>
> **…Generated tests may fail on the code under test while exposing bugs in the code under test…the oracle problem…**
>
> While we agree that the oracle problem exists (and list this as a limitation in Appendix H), we argue it is less significant for our current setup of TestGenEval. We run our benchmark on the “patched” version of SWEBench files, where each of the files has gone through the pull request process and been approved by reviewers. Such code is higher quality than most code on GitHub, and is thus less likely to contain bugs. Our task setup is also still useful in the context of regression testing and also aligns with how developers typically write unit tests. While we might not catch bugs in these files directly, we are still able to catch future regressions with generated tests that obtain high coverage and high mutation score.

---

### Official Review · Reviewer_Yezt · 2024-11-01

**Soundness:** 2
**Presentation:** 3
**Contribution:** 2
**Rating:** 5
**Confidence:** 4

**Summary:**

This paper constructs TestGenEval, a comprehensive and proper benchmark for testing unit test generation abilities of existing LLMs. The benchmark unprecedentedly incorporates mutation scores into their evaluation metric. It also considers multiple generation scenarios including full, first, last, and extra test generation. 10 models including a massive Llama 405B is evaluated on the benchmark and detailed results are given in the main paper and the appendix.
This paper is leaning towards the rejection side since 1. the author did not make effort to evaluate if the generated test matches the definition of a unit test and merely add in mutation score to evaluate further the test effectiveness, and 2. the model selected for evaluation is questionable.

**Strengths:**

- The introduction of mutation score improves on previous benchmarks and ensures a more proper benchmark
- The work introduces the most comprehensive benchmark dedicated to test generation so far
- The work evaluated on a large set of models, even on a very expensive 405B Llama
- Extensive quantitative/correlation analysis, contamination check, and also detailed qualitative analysis, which shows the good quality of the dataset

**Weaknesses:**

### Major
- The evaluation runs the 405B Llama model, which is quite impressive. However, it is unclear why is Llama model evaluated but not other closed-source ones (which are much easier to run comparing to Llama 405B) ones like Claude and Gemini.
- The paper acknowledges that test generated by the LLM can be very different from what human might write. However, there is no procedure in the paper to guarantee that the model generates actually “unit tests” as deemed by human developers (except for the prompt). This means that the model can just generate one big complicated test case for all of the generation scenarios considered (full, first, last, extra), which is against the purpose of unit tests in the first place.
- The task of test generation is generally more solvable by agents or by providing example usages. However, the paper only evaluates on vanilla models and didn’t evaluate even the simple case of adding dependencies into the prompt related to the task.

### Misc
- Citations are not wrapped in brackets (Like Jain et al. when discussing R2E), which is mildly annoying

**Questions:**

- Why is the massive Llama 405B selected in the evaluation rather than other better-performing models like Claude and Gemini?
- What effort does the benchmark made to ensure the test matches the conventional definition of a unit test?
- Why isn't any agents/tools evaluated in the evaluation?

---

> ### Author Response · Authors · 2024-11-17
>
> **Why is the massive Llama 405B selected in the evaluation rather than other better-performing models like Claude and Gemini?**
>
> We chose to evaluate both strong open and closed source models. We wanted to include open-source models as they are more customizable and allow for more privacy, which is important for part of the community. Open source models are also not gated behind paid APIs like Claude and Gemini.
>
> For closed-source models, GPT-4o is widely used and recognized as one of the best models for coding tasks. For open-source models, Llama 405B is a strong model achieving similar performance compared to the most widely-used closed-source models such as GPT-4o, Claude and Gemini. Although Llama 405B is large, we could run it on our servers and were not limited by API costs like with closed-source models.
>
> **What effort does the benchmark made to ensure the test matches the conventional definition of a unit test?**
>
> We follow prior work in generating prompts for unit test generation, and additionally only measure mutation score on the file under test (thus if the model were to write an integration test, it would perform poorly on our benchmark, with low mutation score on the file under test). We filter out tests that do not contain assertions or expect exceptions, marking all such tests as not passing on the code under test. Both our filtering and metrics give us confidence that the model generates tests that match the conventional definition of a unit test. We also perform extensive qualitative analysis, and in all cases we find that the model generates tests that resemble a conventional unit test. We provide a website with all model generations (linked in the paper) if you want to verify our conclusions yourself.
>
>
> **Why isn't any agents/tools evaluated in the evaluation?**
>
> This is one of the limitations we mention in Appendix H. To the best of our knowledge, there is no open agent targeting unit test generation and general agents (e.g. Devin) are not publicly available. We consider that creating our own agents for existing models is out of scope for this paper. We wanted to focus on a simpler setting, where we are able to compare models fairly (with lower variance and susceptibility to agent specific hyperparameters). For those looking to build and run testing agents, we open source all code and dockerized containers for evaluation.

---

### Official Review · Reviewer_g3uj · 2024-11-05

**Soundness:** 3
**Presentation:** 4
**Contribution:** 3
**Rating:** 8
**Confidence:** 5

**Summary:**

This paper introduces a new LLM for test generation benchmark, TestGenEval. It builds on the same repositories as SWEBench, but focuses on extracting the test files and code-test pairs. TestGenEval studies two tasks: test suite generation (generating an entire test suite given a codebase) and test completion (generating test methods to a partial test suite). The generated tests are not only compared against human-written tests based on similarity, but also based on code coverage and mutation score, which are more important metrics for test quality. This paper benchmarked several recent LLMs across different sizes on TestGenEval, and found that generating and completing test suites are challenging tasks for current LLMs.

**Strengths:**

+ Included the measurement of code coverage and mutation score as part of the evaluation metrics, which are important metrics for measuring the quality of generated tests.

+ Conducted a comprehensive set of experiments using LLMs of different sizes and different configurations.

+ Performed an interesting analysis on the current LLMs' performance on TestGenEval, including the correlation with existing benchmarks and common error types.

+ Paper is well-written and easy to follow.

**Weaknesses:**

- Not sure if it is the best to include only the tests appeared in the PRs used by SWEBench; this may miss the other tests that did not appear in those PRs (i.e., don't have recent bug-fixing changes), making the data collection process biased.

**Questions:**

- What is the key difference between the newly created benchmark and the evaluation sets used in prior work on ML for test generation/completion (e.g., Rao et al. 2023, Nie et al. 2023, Dinella et al. 2022, Tufano et al. 2020)? Namely, Rao et al. 2023 had a pretty large dataset for pre-training, and evaluated on the task of generating first/last/additional tests. Both Rao et al. 2023 and Nie et al. 2023 also used runtime metrics (compile and pass).

- line 142, "we next extract code test file pairs from the code and test files run in the PR": do you only consider the tests that appeared in PRs in SWEBench? Or all the tests in those repositories?

- line 189, "This setup is in line with current test completion work Rao et al. (2023); Nie et al. (2023).": this may not be true. The "test completion" in Rao et al. 2023 and Nie et al. 2023 was statement-level completion, but yours is method-level completion, which sounds more like the "test method generation" task in Rao et al. 2023.

- Which mutation tool and mutants did you use? And how many mutants do you generate for each example?

- For test suite generation task, consider renaming "Pass @ 1" to "Any Pass @ 1" to reduce confusion.

- line 361, "the most common error is not generating tests with asserts": but will that lead to a runtime error, and if not, how do you detect them? Also, do you count the generated tests without asserts as pass or not pass?

---

> ### Author Response · Authors · 2024-11-17
>
> **What is the key difference between the newly created benchmark and the evaluation sets used in prior work on ML for test generation/completion?**
>
> Both the scale and quality of TestGenEval are significantly larger than any prior dataset. Dinella et. al. 2022 and Tufano et. al. 2020 leverage the ATLAS and Methods2Test datasets, which focus on mapping test methods to their corresponding method under test (or “focal” method). As a result evaluation is at the method level rather than file level. This is much more simplified than the large scale file level test suite generation and test completion tasks that we target in TestGenEval. Repurposing their datasets would require omitting both of our core tasks.
>
> Nie et. al. and Rao et. al. release test generation datasets that could be repurposed, however their datasets generally consist of simpler programs. Both papers construct their evaluation datasets by trying to build projects and filtering out those that didn’t build, biasing their evaluation datasets towards simpler projects. We provide a table to compare properties of each dataset. TestGenEval has both complex tests and reasonably long code methods, along with more data points than other evaluation datasets. Rao et. al. has longer source method lengths due to increased verbosity of Java code (https://dl.acm.org/doi/pdf/10.1145/3571850), but significantly fewer gold test methods, and lower quality test files (both shorter in length and containing fewer tests on average). The higher number of tests and test length of TestGenEval points to generally higher quality repositories with better testing efforts. We argue these more closely resemble real world software testing. This is also evidenced by the very high star count of repositories used in TestGenEval (3,523-78,287 stars).
>
> | **Paper**       | **Gold Test Methods** | **Average Test Method Len (Tokens)** | **Average Source Method Len (Tokens)** |
> |------------------|-----------------------|--------------------------------------|-----------------------------------------|
> | Nie et al.      | 5,000                 | 86.26                               | 42.85                                  |
> | Rao et al.      | 1,048                 | 64.78                               | 293.24                                 |
> | TestGenEval| 68,647            | 163.37                         | 163.40                             |
>
> Outside of the higher quality of TestGenEval data, TestGenEval is set up in a much more reproducible way compared to prior work. We provide docker images for each file pair and an evaluation harness to compute both coverage and mutation score, enabling the community to easily use and extend our dataset. Building around a widely used dataset such as SWEBench also ensures a degree of data quality, as previous issues with testing inconsistencies and indeterminism have already been figured out by the community.
>
> **Do you only consider the tests that appeared in PRs in SWEBench?**
>
> We consider the code and test files that appeared in PRs in SWEBench, as these files are involved in real world bugs. We argue that testing targeted at these files is more important than testing arbitrary files in a repository, since we are measuring test quality on files that have had historic bugs in them. Furthermore, this paradigm creates a clean mapping between code and test files touched by a patch are directly connected with the code in the patch. This clean mapping ultimately leads to a higher quality dataset that the community can use.
>
> **…yours is method-level completion, which sounds more like the "test method generation" task…**
>
> Our completion task involves generating entire methods for the file to be tested, as Rao et. al. did for their test method generation task. We updated our phrasing in the revised version of the paper to improve clarity.

---

> ### Author Response · Authors · 2024-11-17
>
> **Which mutation tool and mutants did you use? And how many mutants do you generate for each example?**
>
> We used cosmic-ray to generate mutants and use the default set of mutation operators (https://github.com/sixty-north/cosmic-ray/tree/master/src/cosmic_ray/operators). Executing mutants dominates generation in terms of compute cost (our average source file is 1157 LOC and cosmic-ray takes ~15-30 secs per 1000 LOC). For each example, we generate 1031 mutants on average, with approximately 20% of all files we run mutation testing timing out in the one hour allocated for mutation testing execution. However, our error margin for mutation score is low, with an average mutation score uncertainty of 1.06%, meaning our estimates are accurate even in the case mutation testing times out.
>
>
> **…consider renaming "Pass @ 1" to "Any Pass @ 1"...**
>
> We agree that the naming was confusing and updated this in the revised version of the paper.
>
>
> **…not generating tests with asserts…how do you detect them?...do you count…as pass or not pass?**
>
> We detect tests without assertion or exception keywords statically, as not all generated tests will throw a runtime error. We qualitatively also looked at these cases to confirm our keyword match was comprehensive. We mark these cases as not passing to not inflate our metrics.

---

> > ### Comment · Reviewer_g3uj · 2024-12-03
> >
> > Thank you for your response and updates to the paper. They look good to me, and I continue to strongly support the acceptance of this paper.

---

### Author Response · Authors · 2024-11-17

We thank all reviewers for their detailed comments. We tried to address all writing related concerns in our revised version of the paper, with changed text highlighted:
- Clarified how our test completion setting matches the test method generation setting introduced by Rao et. al. 2023
- Changed pass@1 to any pass@1 for our full test suite generation setting
- Refactored our analysis section to include the results from quantitative analysis and reduced qualitative analysis to one example, with the other two examples in appendix
- Clarified that we are looking to generate a test suite for a *file* under test rather than a *class* under test
- Added details on how we generate mutants for mutation testing

---

### Meta-Review · Area_Chair_DQgk · 2024-12-21

**Metareview:**

This paper introduces a new benchmark to support the research on code generation using large language models (LLMs) for real-world software engineering applications. Specifically, the benchmark covers tasks including producing initial test cases, test suite completion, and code coverage improvement, with the goal of comprehensive assessment of test generation performance. The paper evaluates multiple LLMs on this benchmark and finds that even the best-performing models struggles with achieving high code coverage.

The reviewers' were generally positive about the paper, but also raised a number of questions. The author rebuttal answered most questions satisfactorily. One reviewer did not respond to the rebuttal and the corresponding response looks good to me.

Therefore, I recommend accepting the paper and strongly encourage the authors' to incorporate all the discussion in the camera copy to further improve the paper.

**Additional Comments On Reviewer Discussion:**

The author rebuttal answered most questions satisfactorily. One reviewer did not respond to the rebuttal and the corresponding response looks good to me.

---

### Decision · Program_Chairs · 2025-01-22

Accept (Poster)